# Resolving kinetic intermediates during the regulated assembly and disassembly of fusion pores

Debasis Das[1,2], Huan Bao[1,2], Kevin C. Courtney[1], Lanxi Wu[1] & Edwin R. Chapman[1]*

The opening of a fusion pore during exocytosis creates the first aqueous connection between the lumen of a vesicle and the extracellular space. Soluble *N*-ethylmaleimide-sensitive factor attachment protein receptors (SNAREs) mediate the formation of these dynamic structures, and their kinetic transitions are tightly regulated by accessory proteins at the synapse. Here, we utilize two single molecule approaches, nanodisc-based planar bilayer electrophysiology and single-molecule FRET, to address the relationship between SNARE complex assembly and rapid (micro-millisecond) fusion pore transitions, and to define the role of accessory proteins. Synaptotagmin (syt) 1, a major $Ca^{2+}$-sensor for synaptic vesicle exocytosis, drove the formation of an intermediate: committed *trans*-SNARE complexes that form large, stable pores. Once open, these pores could only be closed by the action of the ATPase, NSF. Time-resolved measurements revealed that NSF-mediated pore closure occurred via a complex 'stuttering' mechanism. This simplified system thus reveals the dynamic formation and dissolution of fusion pores.

[1] Department of Neuroscience and Howard Hughes Medical Institute, University of Wisconsin-Madison, 1111 Highland Avenue, Madison, WI 53705, USA. [2] These authors contributed equally: Debasis Das, Huan Bao *email: chapman@wisc.edu

The mechanism by which proteins catalyze the fusion of cellular membranes remain a central unanswered question in cell biology. In all eukaryotic cells, the majority of fusion events are mediated by soluble N-ethylmaleimide-sensitive factor attachment protein receptor (SNARE) proteins[1,2]. The cytoplasmic domains of vesicular SNAREs (v-SNAREs) bind to cognate domains on target membrane SNAREs (t-SNAREs), forming trans-complexes that are necessary and sufficient for fusion[3,4].

Here we focus on the SNAREs that mediate synaptic vesicle (SV) exocytosis in nerve terminals. A key intermediate in this pathway is called the fusion pore, which represents the first aqueous connection between the lumen of a secretory vesicle and the cell exterior[5]. These are nanometer-scale transient structures, lasting only milliseconds before they either close or dilate as the vesicle membrane collapses into the plasmalemma[5,6]. While their structure remains unknown, recent studies indicate that exocytotic pores are composed of both the transmembrane domains (TMDs) of SNARE proteins and phospholipids[7]. Because they are crucial intermediates, fusion pores are a focal point for the action of regulatory proteins[8]. In nerve terminals, fusion pore opening is thought to be triggered by the binding of $Ca^{2+}$ ions to synaptotagmin 1 (syt1)[9,10].

Following complete fusion, v- and t-SNAREs lie within the plasma membrane, in cis-complexes, and must be disassembled to allow for segregation of SNAREs for future rounds of fusion[11,12]. Disassembly occurs when the soluble factor, α-SNAP, binds cis-SNARE complexes and recruits the AAA+ ATPase, NSF; hydrolysis of ATP by NSF drives disassembly. While NSF clearly disassembles cis-SNARE complexes, it remains unclear as to whether the trans-SNARE complex also serves as a substrate[13,14]. This is a crucial question; if trans-SNARE complexes are substrates, then the action of NSF should result in the closure of fusion pores. This would have ramifications in, for example, kiss-and-run exocytosis[15]

The regulated assembly and disassembly of fusion pores have been difficult to study because they are short-lived, so it has been challenging to capture intermediate states. Kinetic analysis of these transitions would provide crucial information concerning the action of regulatory factors that help to assemble, and potentially disassemble, pores. To date, most mechanistic studies of reconstituted fusion machines rely on assays with limited time resolution (i.e., seconds to minutes), thus obscuring dynamics.

To delve into the relationship between the status of trans-SNARE complexes and fusion pore dynamics, we developed a nanodisc (ND)–black lipid membrane (BLM) system[16]. Trans-SNARE pairing results in the formation of individual recombinant fusion pores that can be studied for extended periods because NDs trap pores in intermediate states. The strength of this approach is that pores can be interrogated electrophysiologically[17,18], thus affording microsecond time resolution to reveal kinetic intermediates that could not be previously observed in defined systems.

Here we combine ND-BLM experiments and single-molecule fluorescence resonance energy transfer (smFRET) measurements[19] to determine the impact of key regulatory factors on the structure and kinetic properties of fusion pores and trans-SNARE complexes. The first goal was to assess whether $Ca^{2+}$•syt1 directly regulates SNAREs to trigger pore opening. Indeed, $Ca^{2+}$ and syt1 affected not only the occurrence but also the size and kinetic stability of purely recombinant fusion pores. The observation that $Ca^{2+}$•syt1 dilates pores prompted experiments to address the state of the trans-SNARE complexes that formed them. These experiments demonstrated that $Ca^{2+}$•syt1 mediates the formation of an intermediate: committed trans-SNARE complexes that can only be disassembled by the action of NSF. The time resolution afforded by the ND-BLM system resolved short-lived

fusion pore states and indicated that disassembly occurred via a complex kinetic mechanism. These fusion pore intermediates correspond with trans-SNARE complex status and reveal coupling of SNARE complex assembly with fusion pore transitions.

## Results

**A $Ca^{2+}$ switch converts syt1 from fusion clamp to activator.** In cells, manipulation of a number of syt isoforms alters the occurrence and properties of individual fusion pores[20–23], but it was not clear whether these were direct or indirect effects. To address this question, we used our recently described ND-BLM electrophysiology assay[16] to study pores with microsecond time resolution. We reconstituted full-length syt1, along with syb2, into NDs (Fig. 1a) and t-SNAREs (syntaxin-1A and SNAP-25B heterodimers) into BLMs (Supplementary Fig. 1a); pore formation between the ND and BLM was monitored via the currents that were detected (Fig. 1b). Three kinds of ND preparations were used: $ND3_S$, $ND3_L$, and $ND9_L$, where 3 and 9 refer to the syb2 (and syt1) copy number per ND and subscripts S and L refer to the small (13 nm) and large (30 nm) diameters of the NDs. We first describe representative traces, followed by quantitative analysis of the pores that were formed.

We began with $ND3_S$ and confirmed our earlier observation[16] that these NDs gave rise to pores that remained mostly in the closed state but transiently flickered open to yield small, transient currents; $Ca^{2+}$ had no obvious effect (Fig. 1b). When syt1 was co-reconstituted with syb2 into $ND3_S$, the pore remained mostly in the closed state. However, addition of $Ca^{2+}$ (500 μM $[Ca^{2+}]_{free}$ in all experiments, unless otherwise indicated) resulted in the formation of large, stable pores (Fig. 1b). We note that the properties of individual pores did not differ at the beginning, middle, or end of a recording, but all pores closed within ~90 min. Once the terminal closure occurred, there were no further openings or flickers. Closure might involve reversion to a hemifused state[24]. When syt1 alone was reconstituted (three copies) into NDs (ND0), pores failed to form either in the presence or absence of $Ca^{2+}$ (Supplementary Fig. 1b), confirming that pore formation required trans-SNARE pairing. For the $Ca^{2+}$-free conditions, BAPTA was used to chelate any residual $Ca^{2+}$ present in the buffers; in most of our experiments, we subsequently added $Ca^{2+}$ to yield the indicated $[Ca^{2+}]_{free}$ (note: we confirmed that BAPTA binds $Ca^{2+}$ with a stoichiometry of 1:1 in Supplementary Fig. 1c).

Next, we examined $ND3_L$ (Fig. 1a) but were unable to detect pore formation in the absence of syt1 (Fig. 1c). These finding suggest that v-SNARE density ($ND3_S$, 0.022 syb2/nm², $ND3_L$: 0.0042 syb2/nm²) and not just copy number is a crucial parameter for pore formation. Indeed, a recent modeling study suggested that restricting the mobility of SNAREs, to increase their relative density, facilitates pore formation[25]. In sharp contrast, when syt1 was co-reconstituted into $ND3_L$, robust pore formation occurred upon addition of $Ca^{2+}$ (Fig. 1c); however, these pores exhibited rapid, dramatic flickering behavior.

We then assessed $ND9_L$ (Fig. 1a; 0.013 syb2/nm²) lacking syt1; these NDs yielded pores that remained mostly in the open state but transiently flickered closed (Fig. 1d). In contrast, $ND3_S$, which has a somewhat higher syb2 density, gave rise to pores that were mostly in closed state but transiently flickered open. We conclude that the higher syb2 copy number in $ND9_L$ serves to hold the pore in a stable open state[16]. Strikingly, inclusion of syt1 in $ND9_L$ potently inhibited pore formation in the absence of $Ca^{2+}$ (Fig. 1d). These conditions appear to recapitulate the clamping activity of syt1 that has been observed in neurons[26–29]. Subsequent addition of $Ca^{2+}$ triggered the opening of relatively stable, large pores (Fig. 1d).

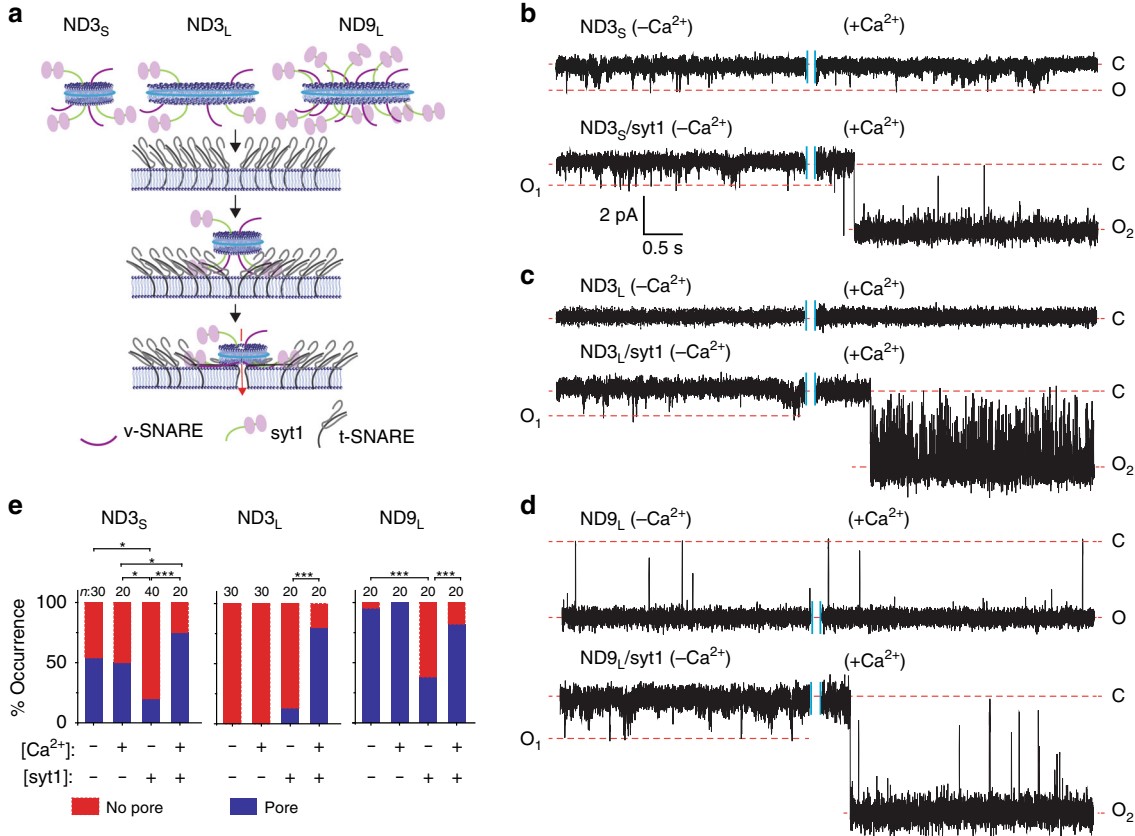

**Fig. 1 Syt1 regulation of single fusion pores measured via planar lipid bilayer electrophysiology. a** Illustration of the nanodisc-black lipid membrane (ND-BLM) system, drawn to scale, indicating the different ND preparations used in this study. The number indicates the syb2/syt1 copy number (1:1) per ND; the subscript S stands for small, 13 nm, NDs and the subscript L stands for large, 30 nm, NDs. Traces of single pores with/without syt1 are shown for ND3$_S$ (**b**), ND3$_L$ (**c**), and ND9$_L$ (**d**). In each trace, minus (−) Ca$^{2+}$ contains 1 mM BAPTA (in all bilayer recording experiments in this study) and plus (+) Ca$^{2+}$ contains 500 μM [Ca$^{2+}$]$_{free}$. Closed (C) and open (O) states are shown; the current and time scale, for all traces, is shown in the inset in **b**. O$_1$ and O$_2$ indicate open state currents obtained before and after addition of Ca$^{2+}$, respectively. **e** Fraction of trials in which a fusion pore was detected, plotted as percentage of occurrence. ND3$_S$, ND3$_L$, and ND9$_L$, in the presence (+) or absence (−) of Ca$^{2+}$ and in the presence (+) and absence (−) of syt1 were compared. Three independent sets of NDs of each type were used, and the total number of measurements obtained under each condition (n) is indicated. Pearson's χ$^2$ analysis of pores formed by ND3$_S$, ND3$_L$, and ND9$_L$ was performed; *p < 0.05, ***p < 0.001.

To conduct quantitative analysis, we carried out 20–40 trials under each of the conditions described above. We first calculated the frequency of pore formation; these findings are represented as "percentage of occurrence" (Fig. 1e). This analysis demonstrated that the occurrence of pores formed by SNAREs alone was unaffected by Ca$^{2+}$ (Fig. 1e). Moreover, inclusion of syt1 in both ND3$_S$ and ND9$_L$ reduced the occurrence of pores in the absence of Ca$^{2+}$ (Fig. 1e). Interestingly, this clamping activity was the most apparent using ND9$_L$, a condition that most closely mimics the syb2 density on SVs (0.014 syb2/nm$^2$)[30]. Using ND3$_L$, we could not address clamping activity because SNAREs alone failed to form pores; under this condition, inclusion of syt1 resulted in the appearance of small, unstable pores (Fig. 1e). In summary, the clamping activity of apo-syt1 is strongly dependent on both SNARE density and copy number (Supplementary Fig. 2). In contrast, Ca$^{2+}$•syt1 clearly increased the occurrence of pores for all three types of NDs that were tested, as compared to the Ca$^{2+}$-free BAPTA control condition.

We next determined whether syt1 affects pore size, as compared to pores formed by SNAREs alone (Fig. 2a); because pores did not form with ND3$_L$ lacking syt1, we focused on ND3$_S$ and ND9$_L$. Pore diameter was estimated via conductance measurements, and these data, from all trials, are plotted in Fig. 2a. Because the length dimension of a fusion pore is not known, and because this parameter might change with tilting of the SNARE TMDs, the pore diameter values, based on conductance measurements, are only approximations. For ND3$_S$ and ND9$_L$ in the absence of syt1 and Ca$^{2+}$, the mean conductance was 170 ± 32 and 563 ± 49 pS, respectively. Addition of Ca$^{2+}$ had little effect (conductance values of 213 ± 56 and 501 ± 47 pS, respectively; Fig. 2a and Supplementary Fig. 3a). Inclusion of syt1 in ND3$_S$ and ND9$_L$ yielded mean pore conductance values of 105 ± 32 and 299 ± 74 pS, respectively, in the absence of Ca$^{2+}$ (Fig. 2a and Supplementary Fig. 3a). Thus apo-syt1 unexpectedly decreases the size of fusion pores (see Supplementary Fig. 3b for estimates of pore size determined for all conditions described in this section; a graphical representation of these data is provided in Supplementary Fig. 13). In contrast, Ca$^{2+}$ greatly increased the size of pores formed by syt1-bearing ND3$_S$ and ND9$_L$, yielding mean conductance values of 415 ± 19 and 902 ± 103 pS, respectively (Fig. 2a and Supplementary Fig. 3a). These results reveal that Ca$^{2+}$•syt1 not only opens pores but also drives dilation (Fig. 2a) at both of the SNARE densities that were tested.

We then compared the kinetic properties of fusion pores formed under various conditions. Open time distribution plots demonstrated that Ca$^{2+}$ had no effect on individual ND3$_S$ pores (Fig. 2b, upper panel); inclusion of syt1 slightly shifted the

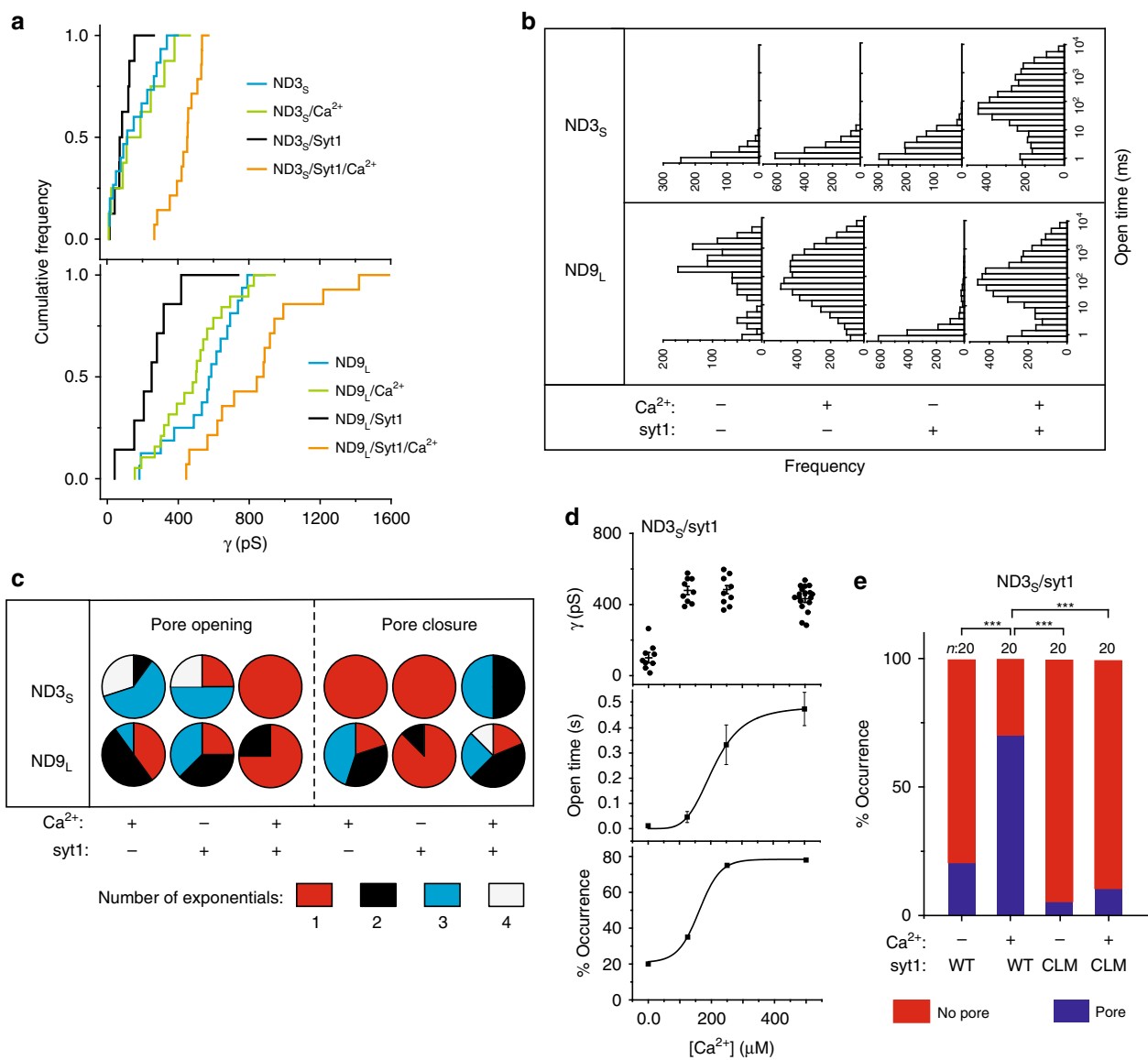

**Fig. 2 Quantification of syt1-regulated fusion pore properties. a** Cumulative distribution functions (CDF) of single-channel conductances across different trials for each experimental condition. **b** Open dwell time histograms for $ND3_S$ and $ND9_L$, plus or minus syt1 and $Ca^{2+}$, are shown. **c** Pie diagrams showing the number of exponentials required to fit the open- and closed-state CDFs for individual pores under the indicated conditions. Details are described in Supplementary Fig. 4. **d** Conductance values ($\gamma$) (upper panel), open time (middle panel), and percentage of occurrence (lower panel) at each indicated $[Ca^{2+}]_{free}$ are plotted. $n = 16, 10, 10, 15$; three different sets of ND preparations were used. In the open time plot, mean values for the open-state dwell times of individual traces were quantified. Error bars indicate SEM. **e** Percentage of occurrence of pore formation, plotted for WT syt1 versus a $Ca^{2+}$ ligand mutant (CLM), D230/232N/D363/365N, that fails to bind $Ca^{2+}$ via either C2 domain. The number of independent trials ($n$) are indicated in **e**. Pearson's $\chi^2$ analysis was performed; ***$p < 0.001$.

distribution to shorter life times in BAPTA while addition of $Ca^{2+}$ drastically increased the open life time (Fig. 2b, upper panel). In the case of $ND9_L$ pores, the findings were more complicated (Fig. 2b, lower panel). Here $Ca^{2+}$ had a small effect on individual $ND9_L$ pores lacking syt1, and inclusion of syt1 shifted the distribution to shorter life times in BAPTA. However, as in the case with $ND3_S$/syt1, addition of $Ca^{2+}$ markedly increased the open life time of $ND9_L$/syt1 pores (Fig. 2b, lower panel). Together, these results demonstrate that syt1 has dramatic, direct effects on the kinetic properties of individual, recombinant, fusion pores. $Ca^{2+}\cdot$syt1 not only increases the size of pores but also promotes membrane fusion by significantly stabilizing the open state.

In order to gain insights into the number of kinetic steps during pore transitions, the cumulative distribution functions

(CDFs) for the opening and closure of individual pores were fitted with single or multiple exponential functions (Supplementary Figs. 4–6). CDFs of the closed and open time distributions reflect the kinetics of pore opening and closure, respectively. In the absence of syt1, pore opening was described by multiple exponentials for both the $ND3_S$ and $ND9_L$ (Fig. 2c); inclusion of apo-syt1 had little effect (Fig. 2c). In sharp contrast, $Ca^{2+}\cdot$syt1 resulted in mainly single exponential kinetics for pore opening, despite the SNARE copy number heterogeneity in our ND preparations (Fig. 2c and Supplementary Figs. 4–6). These findings indicate that $Ca^{2+}\cdot$syt1 drives fusion pores into a single open state with little or no involvement of intermediates.

Regarding pore closure, in the absence of syt1, single exponential kinetics were observed for $ND3_S$, while $ND9_L$ pores followed multi-exponential kinetics. In contrast, inclusion of

apo-syt1 resulted in predominantly single exponential kinetics for both $ND3_S$ and $ND9_L$, even though the number of SNAREs remained the same. Finally, in the presence of syt1 and $Ca^{2+}$, pore closure for $ND3_S$ and $ND9_L$ was best described by multiple exponentials (Fig. 2c and Supplementary Figs. 4–6). These findings indicate that pore closure, in the presence of $Ca^{2+}$•syt1, occurs via a complex multi-state mechanism.

**$Ca^{2+}$ dependencies of fusion pore parameters.** We conducted $[Ca^{2+}]$ dose–response experiments to determine whether the effects of $Ca^{2+}$ on fusion pore conductance, opening, and dilation described in Fig. 2 could be dissociated from one another. Representative traces of individual $ND3_S$/syt1 pores as a function of $[Ca^{2+}]_{free}$ are shown in Supplementary Fig. 7a. The $Ca^{2+}$-dependent effect of syt1 on pore conductance was saturated at $125\,\mu M$ $[Ca^{2+}]_{free}$ (Fig. 2d, top panel); we were unable to reliably determine conductance values at lower $[Ca^{2+}]$, so these data were not fitted to estimate an $EC_{50}$ or Hill slope. The open life time, however, continued to increase at higher $[Ca^{2+}]$, with an $EC_{50}$ of $210 \pm 60\,\mu M$ and a Hill coefficient of $4.3 \pm 0.4$ (Fig. 2d, middle panel). Finally, we determined the fraction of trials in which a pore was observed and found that the $EC_{50}$ was $164 \pm 20\,\mu M$ $Ca^{2+}$ and the Hill coefficient was $3.4 \pm 0.2$ (Fig. 2d, lower panel). The apparent cooperativity in our experiments agree with the Hill slope for release as determined using neurons[31]. We note that all $Ca^{2+}$ titration measurements were carried out using the same $Ca^{2+}$ stocks and buffers. The different $Ca^{2+}$ requirements for each of the fusion pore parameters suggest that different $Ca^{2+}$-binding sites in syt1 (which binds 4-5 $Ca^{2+}$ ions)[9,32,33] might subserve distinct functions during exocytosis.

As a control, we tested a mutant form of syt1 in which two acidic $Ca^{2+}$ ligands, in each C2-domain of syt1, were neutralized by substitution with asparagine residues, thus abolishing $Ca^{2+}$-binding activity[34,35]. This $Ca^{2+}$ ligand mutant was unable to couple $Ca^{2+}$ to fusion pore opening (Fig. 2e).

**Syt1•$PIP_2$ interactions stabilize fusion pores.** It is established that syt1 must penetrate membranes that harbor acidic phospholipids in order to drive membrane fusion in vitro[36] and to trigger exocytosis in cells[26,28,37]. There is evidence that $PIP_2$, which is localized to the plasma membrane, interacts with syt1 under resting conditions to steer the $Ca^{2+}$-triggered membrane penetration of its C2-domains toward the target membrane[38] to regulate fusion in vitro[36]. Indeed, $PIP_2$ plays an essential role in exocytosis in neuroendocrine cells[39,40] and appears to play a key role in at least some modes of SV release[41]. However, before the development of the fully defined ND-BLM system, it had not been possible to directly explore the role of $PIP_2$ on the biophysical properties of individual fusion pores, as previous work relied on cell-based measurements. We therefore carried out ND-BLM measurements using BLMs with and without $PIP_2$. These experiments, using $ND3_S$/$Ca^{2+}$•syt1, revealed that, in the absence of $PIP_2$, pores became less stable (Fig. 3a–c), with the appearance of a partially open state (Fig. 3a). Similar results were obtained using $ND3_L$/$Ca^{2+}$•syt1 but with a more prominent partially open state (Supplementary Fig. 7b, c). As a control, $PIP_2$ had no effect on pores formed by $ND3_S$ in the absence of syt1 (Supplementary Fig. 7d, e). Interesting, $PIP_2$ had a minimal effect on pore occurrence (Supplementary Fig. 7f). As a further control, both PS and $PIP_2$ were omitted from the BLM; under these conditions, $Ca^{2+}$ and syt1 were unable to regulate pores (Supplementary Fig. 8). These experiments confirm that syt1–lipid interactions play a key role in the regulation of fusion pores.

**$Ca^{2+}$•syt1 commits *trans*-SNARE complexes.** We next turned to the question of how syt1 affects *trans*-SNARE complex assembly[42], using a previously described smFRET approach[19]. It was not technically feasible to conduct smFRET using v-SNARE NDs bound to t-SNARE BLMs, so we used v- and t-SNARE NDs. The strength of this approach is that the assembly state of full-length, membrane-embedded SNAREs can be assessed, but we note that the t-SNARE NDs are more constrained as compared to the BLM. Nonetheless, this system revealed that apo- and $Ca^{2+}$-bound syt1 have distinct, direct effects on the assembly of *trans*-SNARE complexes.

Syb2 was labeled with a donor fluorophore (cy3), and syntaxin-1A was labeled with an acceptor (cy5). Monitoring multiple *trans*-SNARE complexes by smFRET is extremely challenging, so we used ND1 for these experiments. For each SNARE, two positions were selected, based on the crystal structure of the *cis*-complex[43,44]: one pair at layer −7 near the N-termini of the SNARE motifs (N-N FRET pair), and the other between layer +4 and +5 near the C-termini of the SNARE motifs (C-C FRET; Supplementary Fig. 9a; Fig. 4). Labeled *trans*-SNARE complexes were first formed using v- and t-SNARE NDs in the absence of syt1. Under these conditions, the FRET ratio ($R$) distribution of the N-terminal pair peaked at ~0.7, while this value was only ~0.2 for the C-C pair; $Ca^{2+}$ had no effect (Fig. 4). Thus single *trans*-SNARE complexes formed between two NDs in the absence of syt1 but exist in a partially zippered state: the N-termini of the SNARE motifs assembled together, but the C-termini remained unzipped. These findings indicate that one SNARE pair is not sufficient to drive fusion in this system, a finding that is also consistent with the inability of $ND3_L$ to form viable pores (Fig. 1e). In the presence of apo-syt1 (reconstituted in the v-ND), the N-N pair remained in the high FRET state, whereas the C-C pair resulted in three different FRET states: 0.2, 0.5, and 0.7 (Fig. 4). These results demonstrate that apo-syt1 drives further assembly of the SNARE complex. We suggest that this structure corresponds to the primed, clamped conformation of the *trans*-SNARE complex. Subsequent addition of $Ca^{2+}$ to these samples had only subtle effects on the smFRET signals (Fig. 4), probably because the C-terminal end of the SNARE motif had largely assembled into a primed and clamped structure[29,45]. However, an increase in the highest C-C FRET peak was apparent, revealing that $Ca^{2+}$-syt1 drives further assembly of the SNARE complex toward the membrane anchors. For completeness, representative raw time-based FRET traces, showing all three FRET states, are shown in Supplementary Fig. 9b. We note that our smFRET experiments are in agreement with previous studies, based on force measurements, that examined the assembly of *trans*-SNARE complexes[46–48] (Supplementary Fig. 9a). In short, the ability of syt1 to clamp membrane fusion prior to the $Ca^{2+}$ signal is associated with its ability to drive assembly of *trans*-SNARE complexes into a more zippered, yet inhibited, state. Then addition of $Ca^{2+}$ drives further assembly and thus fusion pore opening. These findings are consistent with cell-based studies which indicate that the C2B-domain of apo-syt1 is a potent fusion clamp[26,49]. This clamping activity appears to be controlled by conformational changes that determine the relative disposition of the tandem C2-domains. Namely, apo-syt1 clamps fusion when the tandem C2-domains of apo-syt1 are askew[26]. Then, upon binding $Ca^{2+}$, the C2-domains reorient and point in the same direction to trigger exocytosis[26].

In a second approach, we utilized the cytosolic domain of syb2 (cd-syb2), which competes with full-length syb2 in NDs for binding t-SNAREs in the BLM, to prevent or disrupt *trans*-SNARE pairing. In an earlier study, we found that, in the absence of syt1, cd-syb2 readily closes fusion pores, thus revealing that *trans*-SNARE complexes are not fully assembled, or committed,

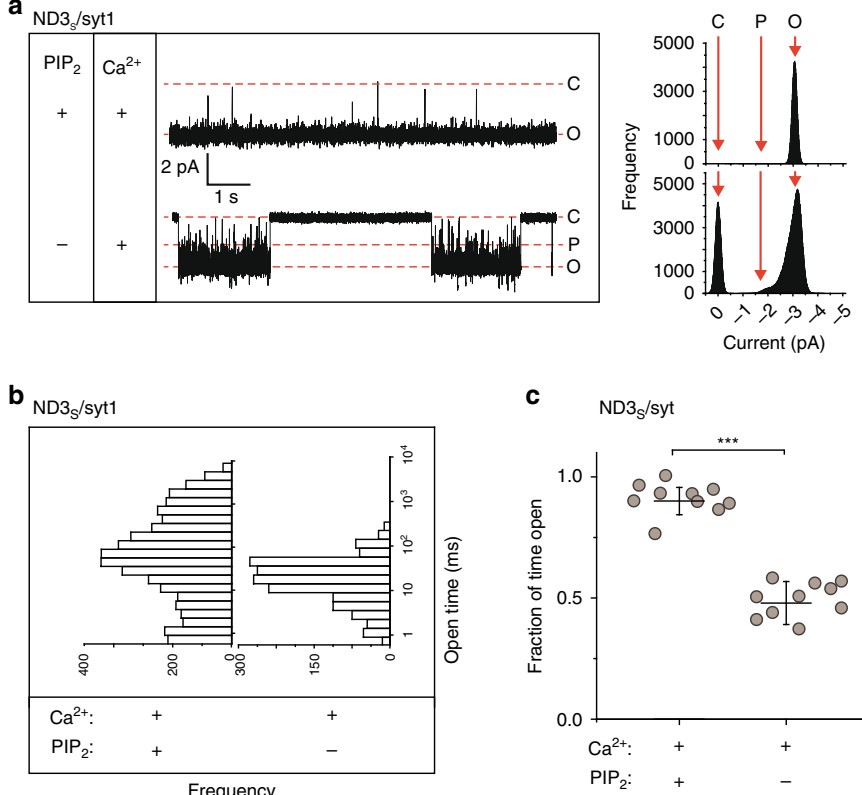

**Fig. 3 Syt1•PIP$_2$ interactions stabilize the open state of fusion pores. a** Traces of single pores, formed using ND3$_S$/syt1 plus Ca$^{2+}$, with (+, upper panel) and without (−, lower panel) 2% PIP$_2$ in the BLM. Respective current histograms are shown beside the traces. Closed (C), open (O), and partially open (P) states are shown; the current and time scale applies to both traces. **b** Open dwell time histogram of pores (combined data from all trials), under the conditions described in **a**. **c** Fraction of time pores were open over a 30-min period, calculated from the data collected under the indicated conditions. Error bars indicate SEM from ten independent BLMs and three independent sets of each kind of ND; the Student's *T* test was performed to compare the two means; ***$p < 0.001$.

at a stage in which fusion pores have opened to a relatively small conductance[16]. Here we started with stable, open ND9$_L$ pores lacking syt1 and, consistent with our previous study, observed that 20 µM cd-syb2 efficiently closed individual pores (Fig. 5a, upper panel); Ca$^{2+}$ was without effect. We then carried out the same experiments using ND9$_L$/syt1. After the pore was opened by Ca$^{2+}$ and robust currents, indicative of large pores, were detected, 20 µM cd-syb2 was added. Remarkably, pore closure was not observed, even over a 45-min recording period (Fig. 5a, lower panel). We then repeated this experiment but first added excess BAPTA to chelate Ca$^{2+}$ and deactivate syt1; even then, subsequent addition of cd-syb2 was without effect (Fig. 5b and Supplementary Fig. 10a, b). Thus the action of Ca$^{2+}$•syt1 appears to be terminal, as the ongoing presence of Ca$^{2+}$ is not required to maintain pore properties.

To formalize these observations, we calculated the fraction of time that individual pores were in the open state over a 45-min period. This fraction decreases in the presence of cd-syb2 when SNAREs alone were used (Fig. 5c). In contrast, after the action of Ca$^{2+}$•syt1, the fraction open was unaffected by cd-syb2 (Fig. 5c). These results further demonstrate that syt1 drives assembly of *trans*-SNARE complexes such that they enter into a committed, in effect "irreversible," state that requires additional energy to become disassembled.

**α-SNAP/NSF-ATP closes fusion pores**. Since fusion pores formed in the presence of Ca$^{2+}$ and syt1 comprise committed *trans*-SNARE complexes, we examined whether they could be closed by NSF, an AAA+ ATPase that acts to disassemble

SNARE complexes in vitro[11] and in vivo[50]. NSF is recruited to SNARE complexes via an adaptor protein, α-SNAP[11,12]. Then, upon ATP hydrolysis by NSF, SNARE complexes are disassembled[11,51]. In vivo, NSF is thought to act on *cis*-SNARE complexes in the plasma membrane following fusion, where disassembly allows v- and t-SNAREs to be segregated into separate compartments for subsequent rounds of fusion. Whether NSF can act on *trans*-SNARE complexes has been the subject of debate[13,14]. The ND-BLM assay makes it possible to determine directly whether NSF acts on *trans*-SNARE complexes in a functional manner, as disassembly would be evidenced by pore closure. As shown in Fig. 6, addition of α-SNAP/NSF-ATP to fusion pores formed by ND3$_S$/syt1•Ca$^{2+}$ resulted in dramatic changes: rather than brief flickers to the closed state, prolonged closures became apparent, eventually leading to complete, irreversible closure. We note that a distinct subconductance state appeared in 67% of open-to-closed and 75% of closed-to-open transitions during NSF-mediated pore dissolution (Fig. 6a, b and Supplementary Fig. 11a). In all five trials, we observed repeated transitions between open/partially open and closed states before the final, irreversible, closure. We term this a "stuttering" mechanism for pore dissolution. Apparently, there are intermediate steps that are reversible during disassembly. Irreversible closure likely results from loss of SNAP-25B, which becomes too dilute after disassembly to efficiently re-bind syntaxin-1A. Use of the ATP analog ATP-γ-S (Fig. 6a, lower panel, 6a) or omission of Mg$^{2+}$ (Supplementary Fig. 11b) abolished the action of α-SNAP/NSF-ATP on pore closure, confirming that disassembly depended on efficient hydrolysis of ATP. Finally, the individual

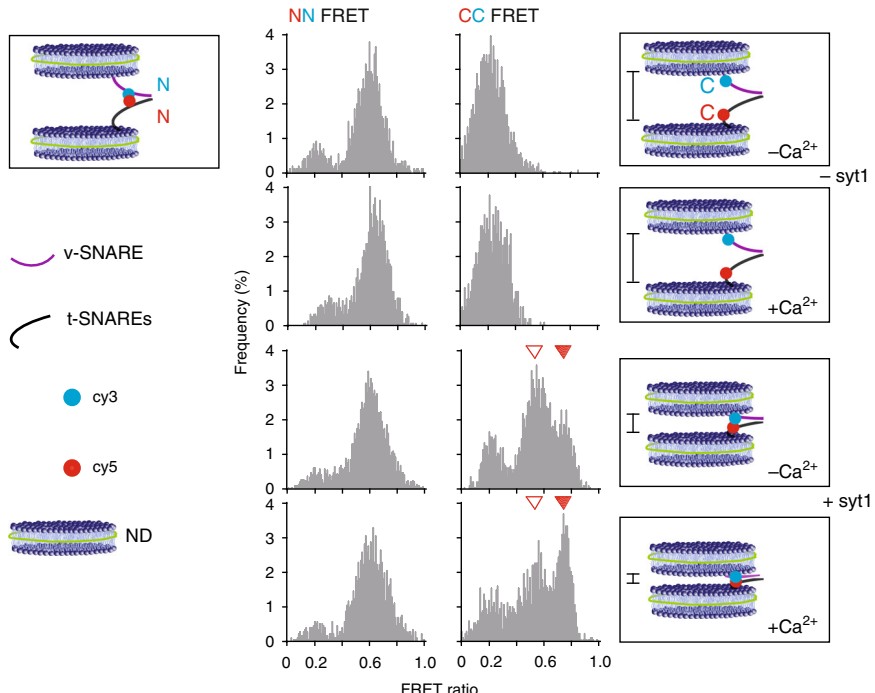

**Fig. 4 smFRET reveals the impact of syt1 on *trans*-SNARE complex assembly.** smFRET histograms of *trans*-SNARE complexes, under the conditions indicated on the far right, using FRET pairs (cy3, shown as blue dots and cy5, shown as red dots) near the N- (NN; left panels) or the C-termini (CC; right panels) of the SNARE motifs of syb2 and syntaxin-1A, respectively. For clarity, the t-SNARE heterodimer is presented by a single black line, and because NN smFRET was not affected by $Ca^{2+}$ or syt1, only the starting condition is illustrated for the NN experiments (upper left). In the illustrations on the right (+/− $Ca^{2+}$ and syt1), vertical lines indicate the relative distance between v- and t-SNARE NDs under each condition. NN: $n = 67$ (−$Ca^{2+}$, −syt1), 61 (+$Ca^{2+}$, −syt1), 83 (−$Ca^{2+}$, +syt1), 64 (+$Ca^{2+}$, +syt1). CC: $n = 79$ (−$Ca^{2+}$, −syt1), 69 (+$Ca^{2+}$, −syt1), 74 (−$Ca^{2+}$, +syt1), 87 (+$Ca^{2+}$, +syt1). ND1$_S$ were used in all experiments. Data were collected using three independent sets of NDs. In all smFRET experiments, −$Ca^{2+}$ samples contained 1 mM EGTA; +$Ca^{2+}$ samples contained 500 μM $[Ca^{2+}]_{free}$.

components: α-SNAP, NSF-ATP, or $MgCl_2$, had no appreciable effect on pores, demonstrating that all disassembly factors must be present at the same time in order to close pores (Supplementary Fig. 11c). Together, these results demonstrate that α-SNAP/NSF can close fusion pores, presumably by disassembly of committed *trans*-SNARE complexes, and this effect requires ATP hydrolysis by NSF (Fig. 6c). In summary, syt1 acts to promote SNARE complex assembly to the point where it requires additional energy, provided by the ATPase NSF, to disassemble.

## Discussion

At present, little is known concerning the structure and dynamics of *trans*-SNARE complexes because they have been difficult to trap in distinct functional states. Most of what is known regarding SNARE complex assembly and disassembly stems from studies based on *cis* complexes[43,44,52], which do not form until after fusion. The major goal of the current study was to address the relationship between *trans*-SNARE complex assembly and fusion pore properties. We approached this question by determining how transitions in these structures are affected by two classes of regulatory proteins: one—syt1—is thought to trigger pore opening, while the other—α-SNAP/NSF—serves to disassemble SNARE complexes. To study kinetic transitions in pores, we applied a newly described ND-BLM method, which affords microsecond time resolution. This approach was combined with smFRET measurements to monitor SNARE zippering.

The first goal was to address the impact of syt1 on recombinant fusion pores. Under resting conditions, it has been proposed that apo-syt1 serves as a fusion clamp that prevents SV exocytosis until the arrival of an action potential and a concomitant increase in $[Ca^{2+}]_i$[26,28,29,49]. We observed that a clamping function for

syt1 emerged in the ND-BLM system and was most prominent when fusion pores were formed using physiological densities of syb2[30] (Figs. 1 and 2). smFRET measurements indicate that the ability of syt1 to clamp fusion, under resting conditions, was mediated by a direct action on *trans*-SNARE complexes (Fig. 4). In the absence of $Ca^{2+}$, spontaneous openings still occurred to some degree, and kinetic analysis revealed that these openings followed multi-exponential kinetics (Fig. 2c), indicating the involvement of multiple intermediates. In contrast, in the presence of both $Ca^{2+}$ and syt1, pore opening followed single exponential kinetics, suggesting that all *trans*-SNARE complexes were driven into the same primed state, making the pore opening the result of a single collective stroke. It will be interesting to determine which of the distinct syt1-SNARE complex structures that have been reported[45,53,54] coincide with the primed but clamped state, and with the fusogenic state, of this protein complex.

Syt1 is unlikely to clamp fusion by preventing the docking of NDs to the BLM via steric effects, as this protein has been reported to facilitate docking in reconstituted systems and in synapses[36,55]. Moreover, we used physiologically relevant syb2 and syt1 densities that approximate the densities found on native SVs[30]. Indeed, SVs contain a myriad of additional proteins, so are more crowded than our NDs. Finally, the observation that syt1 inhibits pore activity, at least in part, by reducing the open life time in a dose-dependent manner (Supplementary Fig. 2) indicates that aspects of the inhibitory/clamping activity of syt1 are mediated by the intrinsic properties of the protein.

Upon addition of $Ca^{2+}$, syt1 efficiently triggered fusion pore opening in the ND-BLM system, under all conditions that were tested. So we conducted experiments to probe for changes in the

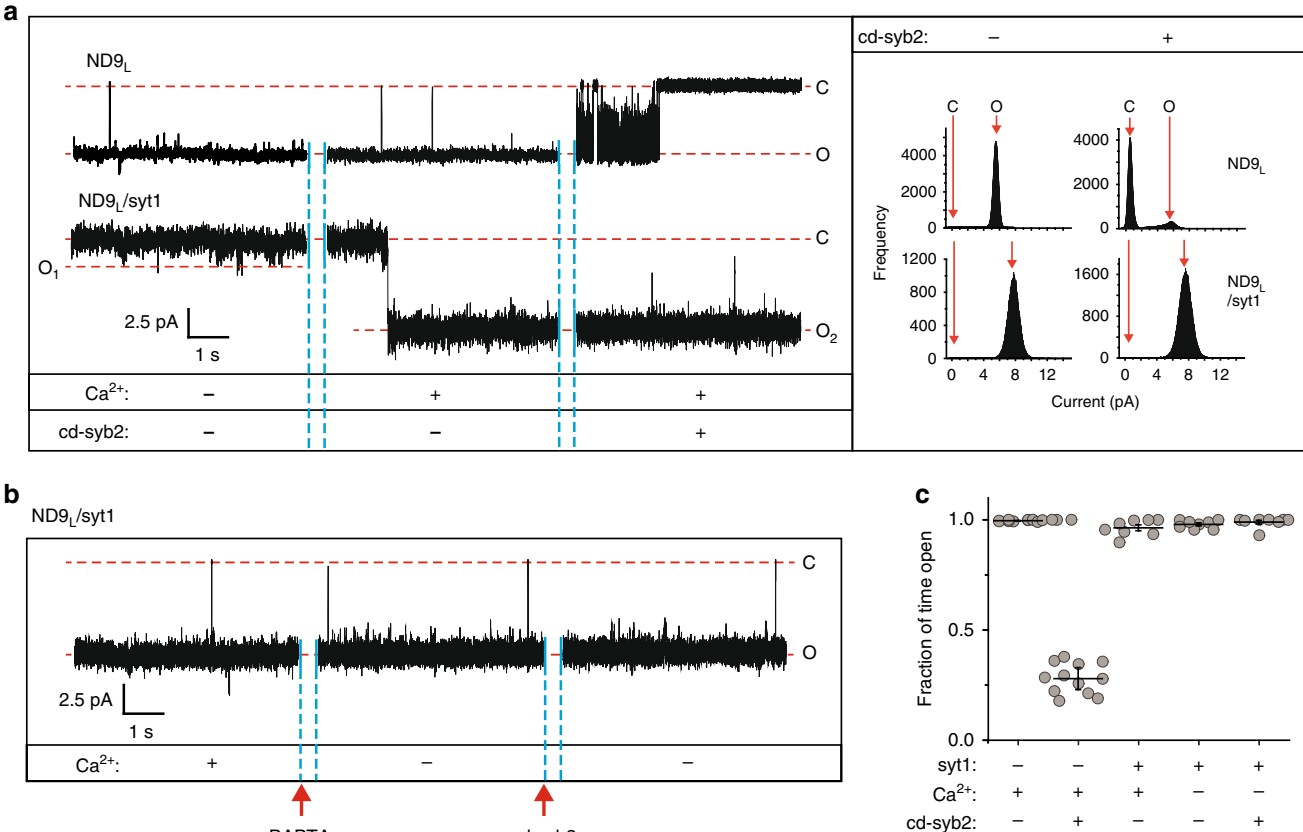

**Fig. 5 Ca²⁺•syt1 drives *trans*-SNARE complexes into a functionally committed state. a** Representative recording of ND9$_L$ lacking or containing syt1. Once pores were open, 20 μM cd-syb2 was added to each reaction. Closed (C) and open (O) states are indicated; the current/time scale for all traces is shown in the inset, left. Right panel: respective current histograms are shown. **b** Representative recordings of ND9$_L$ bearing syt1 in 500 μM [Ca²⁺]$_{free}$ are shown, followed by the addition of 1.5 mM BAPTA (arrow), further followed by 20 μM cd-syb2 (arrow). The current/time scale is shown on the bottom left. **c** Fraction of time individual pores were open over a 45-min period, plotted for each condition as indicated. Error bars indicate SEM from 12 and 8 independent BLMs for **a** and **b**, respectively; 3 independent sets of NDs were used.

structure of the underlying SNARE complexes. We previously observed, using SNAREs alone, that addition of cd-syb2 resulted in the disassembly of *trans*-SNARE complexes and pore closure[16]. These findings prompt the question of how far pores must progress such that SNAREs assemble into highly stable state that requires the action of NSF in order to be disassembled. Indeed, Ca²⁺•syt1 not only opened pores but also drove pore dilation (Supplementary Fig. 12), and this was associated with further assembly of *trans*-SNARE pairs as revealed by smFRET. Remarkably, once pores were opened by Ca²⁺•syt1, they became resistant not only to a Ca²⁺ chelator but also to high concentrations of cd-syb2. Apparently, *trans*-SNARE complexes had entered into a committed state such that cd-syb2 could no longer displace membrane-anchored syb2, within SNARE complexes, even over relatively long time frames (45 min). Committed fusion pores still flickered, and this might reflect their partially lipidic structure[7,56,57]. We also note that the interaction of syt1 with PIP₂[36,38], a plasma membrane lipid that plays a key role in exocytosis, served to stabilize the Ca²⁺-triggered opening of fusion pores, so lipids also impact—directly or indirectly—fusion pore transitions.

Given that fusion pores triggered to open by the action of Ca²⁺•syt1 were functionally committed, we asked whether α-SNAP/NSF/ATP was able to disassemble the SNARE complexes that formed them. While it is well established that the action of α-SNAP and NSF serves to disassemble *cis*-SNARE complexes after fusion, whether these factors can disassemble

*trans*-complexes remains unresolved[13,14]. The ND-BLM system provides a functional read-out that relates SNARE complex disassembly with fusion pore transitions in real time. Indeed, upon hydrolysis of ATP, NSF and α-SNAP did in fact close fusion pores formed by *trans*-SNARE complexes. This experiment further demonstrates that SNARE zippering and unzippering reactions underlie structural and kinetic transitions in individual fusion pores. Moreover, these findings support the conclusion that we are studying *trans*, and not *cis*, complexes, as disassembly of *cis* complexes would not be expected to close pores. It is possible that the TMD of syb2 binds to the MSP, preventing zippering with the TMD of syntaxin[43], but we did not observe binding in pull-down assays (Supplementary Fig. 13). However, since these experiments were performed in the presence of detergent, it remains formally possible that interactions might occur within lipid-filled NDs. It is also possible that the curvature of the membranes, or the proteinaceous aspects of the pore, prevent the v- and t-SNARE TMDs from coalescing.

In all trials, NSF-mediated closure involved flickering behavior and the appearance of a subconductance state. This state was most commonly observed during pore transitions (Fig. 6a and Supplementary Fig. 11a). Because the partially open state was associated with both open-to-closed, as well as closed-to-open, transitions, it is not simply a product of vectorial disassembly. In this light, we note that cd-syb2 can give rise to subconductance states when used to close pores formed by SNAREs alone (Supplementary in ref. [16]); this occurs via the formation of partially assembled

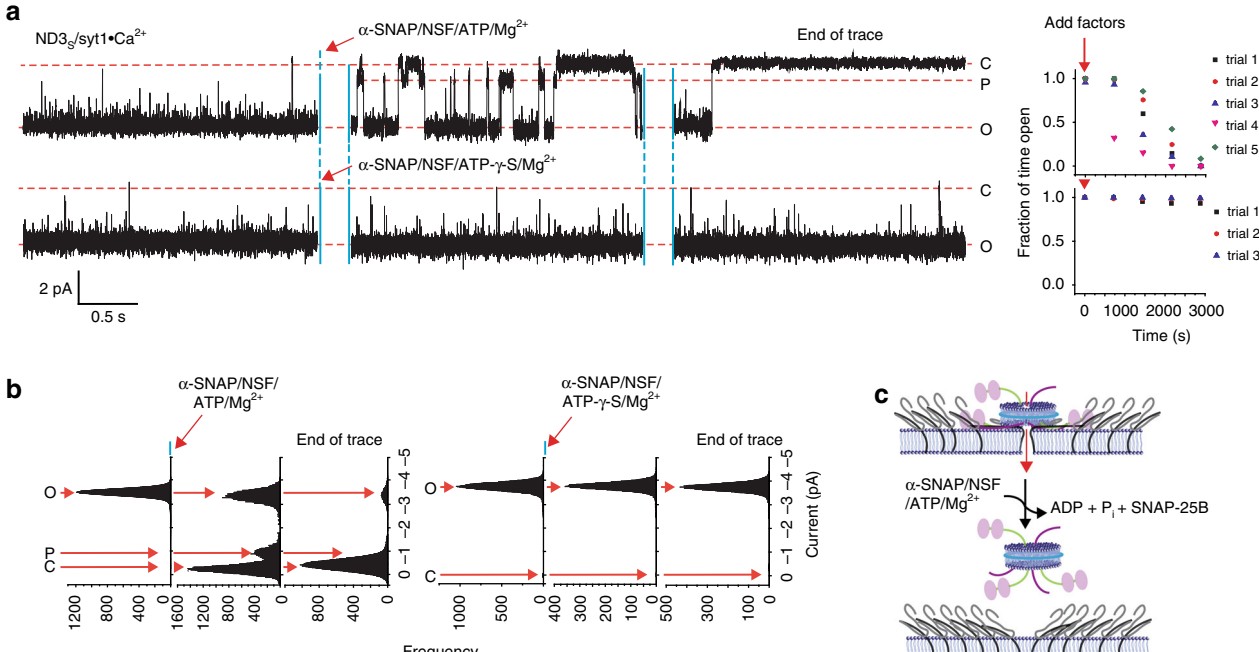

**Fig. 6 α-SNAP/NSF-ATP disassemble *trans*-SNARE complexes formed by Ca$^{2+}$•syt1. a** Representative recording of a pore formed by ND3$_S$/syt1 in 500 μM [Ca$^{2+}$]$_{free}$. After pore formation, 0.3 μM α-SNAP, 0.3 μM NSF, 1 mM ATP, and 5 mM MgCl$_2$ were added (red arrow). In the lower panel, ATP was replaced with 1 mM ATP-γ-S. Two different epochs of each trace are shown; the time interval between each epoch is indicated. Closed (C), open (O), and partially open (P) states are marked; the current/time scale for all traces is provided in the inset. After addition of the disassembly factors, recordings were divided arbitrarily into five 720-s epochs; the fraction of time the pores were open during each epoch was quantified for individual traces and plotted beside each panel. *n* = 5 BLMs using ATP, and *n* = 3 BLMs using ATP-γ-S; two different sets of NDs were used for each condition. **b** Representative current histograms for the experiments described above are shown. **c** Illustration (drawn to scale) shows the α-SNAP/NSF/ATP/Mg$^{2+}$-mediated disassembly of SNAREs, leading to fusion pore closure.

SNARE complex intermediates. It is likely that α-SNAP/NSF acts in a similar manner, by allowing the forward formation of partially assembled *trans*-SNARE complexes, during ongoing disassembly, to yield a subconductance state. According to this view, there is a tug-of-war in which disassembly and assembly oppose one another, even during the ongoing action of NSF. One model to explain this tug-of-war is that the fusion pore is formed by two SNARE complexes. The action of NSF on one SNARE pair would close the pore, but the other SNARE complex would hold the ND in place so that the partially or completely disassembled pair can reassemble and thus re-open the pore. In this model, complete closure would occur when both SNARE pairs are disassembled at the same time. Regardless of the detailed underlying model, these time-resolved measurements indicate that functional disassembly of a pore is a complex process. Moreover, our findings raise the possibility that this disassembly reaction might underlie aspects of kiss-and-run exocytosis[8,15]. Interestingly, binding of SM (Sec1-Munc18) proteins in synapses[58] or the binding of HOPS (Homotypic fusion and protein sorting) complex in case of vacuolar fusion[59] might protect *trans*-SNARE complexes from NSF/α-SNAP-mediated disassembly[60]. Future studies will reveal whether these factors affect NSF-mediated fusion pore closure in the ND-BLM system described here.

In summary, we report key kinetic intermediates that are formed during the regulated assembly and disassembly of fusion pores. During assembly, the major Ca$^{2+}$ sensor for SV exocytosis, syt1, directly regulates the occurrence, size, and dynamics of fusion pores by promoting SNARE zippering into a functionally committed state. This committed state could only be disassembled upon ATP hydrolysis by NSF, resulting in pore closure. Thus the defined, reduced, reconstituted ND-BLM system

makes it possible to measure, in real time, the entire SNARE cycle, while revealing the behavior of fusion pores as they are constructed and deconstructed.

## Methods

**Materials.** 1-Palmitoyl-2-oleoyl-sn-glycero-3-phosphocholine (PC), 1,2-dioleoyl-sn-glycero-3-phospho-l-serine (PS), 1,2-dioleoyl-sn-glycero-3-phosphoethanolamine (PE), 1,2-diphytanoyl-sn-glycero-3-phosphocholine (DPhPC), 1-palmitoyl-2-oleoyl-sn-glycero-3-phospho-(1'-rac-glycerol) (sodium salt) (PG), and brain PI (4,5)P2 were purchased from Avanti polar lipids; DDM (n-dodecyl β-D-maltoside) and OG (n-octyl glucoside) were from Gold Biotechnology; IPTG was from Research Products International Corp.; Triton X-100 was from Sigma Aldrich; Ni-Sepharose 6 Fast Flow was from GE Healthcare; 2-mercaptoethanol and glycerol were from Thermo Fischer Sc.; Bio-beads SM2 were from BIO-RAD; ATP-γ-S was from Abcam.

**Protein purification.** All cDNA used in this study was derived from rat. Syb2, t-SNARE heterodimers comprising syntaxin-1A and SNAP-25B, and full-length syt1 were expressed and purified as his$_6$-tagged proteins, as described previously[7,36]. Syb2 and t-SNAREs were expressed at 37 °C in *Escherichia coli* BL21 (DE3) cells (ThermoFisher Scientific, Catalog number: C600003), whereas syt1 was expressed at 28 °C. In brief, bacterial pellets were resuspended (~10 ml per liter of culture) in resuspension buffer (25 mM HEPES-KOH [pH 7.4], 400 mM KCl, 10 mM imidazole, and 5 mM β-mercaptoethanol) and incubated for 20 min on ice after addition of 0.5 mg/ml lysozyme. Protease inhibitor (1 mM PMSF), DNase I, and RNase (Sigma, 10 μg/ml) were then added and samples were sonicated in 35 ml batches on ice for 2× 45 s (50% duty cycle). Triton X-100 was added to 2.1% (v/v) and incubated overnight with rotation at 4 °C before centrifugation of the cell lysate at 19,000 × g for 30 min in a JA-17 rotor (Beckman). The supernatant was then incubated for >2 h at 4 °C with Ni-NTA agarose (Qiagen; 0.5 ml of a 50% slurry per liter of cell culture) equilibrated in resuspension buffer. Beads were washed extensively with resuspension buffer containing 1% Triton X-100 and then washed with OG wash buffer (25 mM HEPES-KOH [pH 7.4], 400 mM KCl, 50 mM imidazole, 10% glycerol, 5 mM βmercaptoethanol, 1% octyl glucoside). The slurry was loaded onto a column, washed with 5–10 column volumes of OG wash buffer, and step-eluted with OG wash buffer containing 500 mM imidazole.

NSF, α-SNAP, the cytoplasmic domain of syb2, and membrane scaffold proteins (MSP1E3D1 and NW30) were also purified as his6-tagged proteins, as described previously[12,16,61,62]. In brief, a similar procedure as above was used to purify these proteins except all detergents were omitted from the wash buffers. The purified proteins were dialyzed against 25 mM HEPES-KOH (pH 7.4), 100 mM KCl, 10% glycerol, and 1 mM dithiothreitol (DTT).

**Proteoliposome reconstitution**. t-SNARE liposomes were prepared as described previously. In brief, t-SNARE heterodimers were mixed together with lipids (25% PE, 75% PG) in reconstitution buffer (25 mM HEPES, pH 7.5, 100 mM KCl, and 1 mM DTT) plus 0.02% DDM. Detergent was removed with BioBeads under gentle shaking (overnight, 4 °C). t-SNARE liposomes were then isolated by flotation[16], followed by dialysis against reconstitution buffer (overnight, 4 °C).

**ND reconstitution**. Reconstitution of syb2 into NDs was performed as described[7,62]. In some experiments, full-length recombinant syt1[36] was co-reconstituted with syb2 at a 1:1 molar ratio unless otherwise indicated. MSP1E3D1 was used to generate small, 13 nm, NDs (ND$_S$) and NW30 was used to prepare large, 30 nm, NDs (ND$_L$). The ratios of MSP to lipid molecules were 2:120 for ND$_S$ and 2:1000 for ND$_L$; the MSP-to-syb2 ratios were 2:1 (ND3) and 2:8 (ND9). The copy number of syb2 and syt1 per ND refers to the total number of syb2 and syt1 molecules, not the number of copies per face of the ND. The lipid composition was 40% PS, 45% PC, and 15% PE. Briefly, reconstitution involved mixing syb2, MSP, and lipids, with or without syt1 in reconstitution buffer containing 0.02% DDM. Detergent was slowly removed with BioBeads (1/3 volume, BIORAD) with gentle shaking (overnight, 4 °C). The preparation was centrifuged (20 min at 100,000 × g) to remove aggregates, and the NDs in the supernatant were purified by gel filtration using a Superdex 200 10/300 GL column, equilibrated in reconstitution buffer plus 5% glycerol.

**Planar lipid bilayer electrophysiology**. Planar lipid bilayer recordings were performed using a Planar Lipid Bilayer Workstation from Warner Instruments (USA) as described[16,63,64]. Briefly, lipids (30% DOPE, 52% DPhPC, 16% DOPS and 2% brain PIP$_2$, at 30 mg/ml in n-decane) were first painted onto a 150-μm aperture in a 1-ml, white Delrin or polystyrene cup (Warner Instruments) and dried for 15 min. Then the aperture was bathed in 1 ml of 25 mM HEPES, pH 7.4 with 100 mM KCl in the cis and 10 mM KCl in the trans chamber. The lipid solution was gently re-applied to the hole until a conductance-blocking seal was formed, as determined by capacitance measurements. This process was repeated, either with a brush or air bubble, until the desired capacitance was achieved.

**Single-channel measurements of fusion pores**. After BLM formation, t-SNARE proteoliposomes (75% PE and 25% PG) were added to the cis chamber of the apparatus; these spontaneously fuse with the planar bilayer, thus depositing the t-SNAREs into the BLM. t-SNAREs were reconstituted into BLMs, at a density of 0.4 molecules per μm$^2$[16]. In our experiments, ~40 t-SNARE vesicles (of 40 to 50 nm diameter) fuse with the BLM[16], altering the lipid composition of the BLM by only 0.0018%. Then, to form fusion pores, v-SNARE NDs (reconstituted with/without syt1) were added to the cis chamber. Pores form within 10–40 min and could be monitored for >90 min, before terminal closure. Currents were recorded using Bilayer Clamp Amplifier BC-535 (Warner Instrument) and a Digidata 1550B (with Humsilencer) acquisition system (Molecular Devices Corp.). Single-channel recordings were acquired at 10 kHz using the pCLAMP 10 (Molecular Devices, LLC.) software and were filtered at 5 kHz using a multisection Bessel filter. $\Delta\psi \equiv \psi_{cis} - \psi_{trans}$ ($\psi_{trans} \equiv 0$ V). All recordings were conducted at room temperature.

Experiments were initiated with 1 mM BAPTA in the cis chamber, followed by the sequential addition of reconstituted liposomes bearing t-SNAREs and then NDs. This was followed by addition of 1.5 mM CaCl$_2$ in the cis-chamber to yield 500 μM [Ca$^{2+}$]$_{free}$. To generate dose–response curves, different concentrations of CaCl$_2$ were added to yield the indicated [Ca$^{2+}$]$_{free}$ in Fig. 2d and Supplementary Fig. 7a. Pore formation and dynamics were studied at $\Delta\psi = -60$ mV.

To study the effect of PIP$_2$ on fusion pores in Fig. 3 and Supplementary Fig. 7b–f, the lipid composition of t-SNARE proteoliposomes and NDs were kept constant, but the BLM lipid composition was varied as indicated in the text. BLMs were formed following same technique as described above.

To study the effect of cd-syb2, BAPTA, α-SNAP, NSF, ATP, and Mg$^{2+}$, the indicated concentration of each factor was added to the cis-chamber, after pores had opened.

**Single-channel analysis of fusion pore unitary currents**. All single-channel data were analyzed using Clampfit 10.7 (Molecular Devices, LLC.) and MS Origin 2016 (OriginLab, USA). In all figures showing BLM recordings, the representative traces were filtered at 1 kHz for display purposes.

Current histograms were plotted using CLAMPFIT 10.7 and fitted with Gaussian functions. To calculate the open life time of individual pores, 30-min periods (unless otherwise indicated) from individual traces were analyzed. Dwell times corresponding to the fully open and fully closed states were measured in

individual records using the event detector in CLAMPFIT10.7. Open-state dwell time histograms were plotted using MS Origin 2016 (OriginLab, USA).

The fraction of time fully open was calculated using the equation:

$$\frac{\text{Open dwell time}}{\text{Open dwell time} + \text{closed dwell time}} \qquad (1)$$

**Kinetic analysis of single-channel data**. Single-channel kinetic analysis was performed using previously reported methods[65]. Closed-state ($t_{closed}$) and open-state ($t_{open}$) dwell times detected in CLAMPFIT10.7 were statistically analyzed by CDFs [defined by the probability $P(t_{closed} \leq t)$ and $P(t_{open} \leq t)$] using Origin 2016. Closed- and open-state CDFs of individual traces were fitted separately; representative CDFs with the different exponential fits are shown in Supplementary Fig. 4. CDFs generated from the closed- and open-state dwell times provided kinetics for the pore opening and pore closure, respectively. Exponential equations used to fit the CDFs are provided here.

$$\text{One exponential function} : \text{CDF} = 1 - \exp(-kt). \qquad (2)$$

$$\text{Two exponential function} : \text{CDF} = 1 - A_1\exp(-k_1 t) - A_2\exp(-k_2 t). \qquad (3)$$

$$\text{Three exponential function} : \text{CDF} = 1 - A_1\exp(-k_1 t) - A_2\exp(-k_2 t) \\ - A_3\exp(-k_3 t). \qquad (4)$$

$$\text{Four exponential function} : \text{CDF} = 1 - A_1\exp(-k_1 t) - A_2\exp(-k_2 t) \\ - A_3\exp(-k_3 t) - A_4\exp(-k_4 t). \qquad (5)$$

$$\text{Five exponential function} : \text{CDF} = 1 - A_1\exp(-k_1 t) - A_2\exp(-k_2 t) - A_3\exp(-k_3 t) \\ - A_4\exp(-k_4 t) - A_5\exp(-k_5 t). \qquad (6)$$

Akaike Information Criterion was used to determine the goodness of fit[65,66];

$$\text{AIC} = n\log(\text{RSS}/n) + 2(k + 1). \qquad (7)$$

RSS in Eq. 7 is the residual sum of squares, $n$ is the number of data points, and $k$ is the number of parameters in each model tested.

**Single-molecule FRET measurements**. Flow cells and 13 nm NDs bearing labeled SNAREs for single-molecule experiments were prepared as described previously[19,67]. Native cys residues were replaced with serines, and then cys residues were introduced for chemical labeling: syb2 was labeled at positions 33 or 72 with cy3 and syntaxin-1A was labeled at positions 203 or 241 with cy5. v- and t-SNARE NDs (5 μM) were incubated at room temperature for 30 min in the reconstitution buffer supplemented with 0.5 mM EGTA or Ca$^{2+}$. This is similar to the waiting time for fusion pore formation in the ND-BLM system (10–30 min). Samples were then diluted to 10 pM before injection into flow cells. Unbound NDs were washed and samples were imaged in imaging buffer consisting of (in mM): 1 Trolox, 0.5 CaCl$_2$, 100 KCl, 25 HEPES, and an oxygen scavenging system (1% glucose, 1 mg/ml glucose oxidase, and 0.02 mg/ml catalase), pH 7.4. For experiments carried out in the absence of Ca$^{2+}$, samples included 0.5 mM EGTA. Single-molecule imaging experiments were performed using a prism-based total internal reflection fluorescence microscope with 100-ms time resolution[19]. The FRET ratio ($R$) was calculated using the following equation:

$$R = \frac{\text{IA} - 0.05\text{ID}}{\text{ID} + \text{IA}} \qquad (8)$$

In Eq. 8, ID and IA are the donor and acceptor intensities after background subtraction. Single-molecule spots were identified by alternating the 532 and 653 nm laser excitation to confirm the presence of both v- and t-SNARE NDs. Traces showing a single photobleaching event were selected for analysis. Each experiment was repeated at least three times independently, and one data set for each condition is presented. smFRET data were processed using custom MATLAB scripts. We fitted the FRET histograms with two- or three-component Gaussians to derive the FRET ratio ($R$) for each state.

**Statistical analysis**. The number of independent trials is provided in the figure legends, along with the statistical tests that were performed. Error bars represent SEM.

## Data availability
Data supporting the findings of this manuscript are available from the corresponding author upon reasonable request. A reporting summary for this article is available as a Supplementary Information file. The source data underlying Figs. 1e, 2a, c, e, 3c, and 5c and Supplementary Figs. 3a, b, 7f, and 11a are provided as a Source Data file.

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

## Acknowledgements

We thank Meyer B. Jackson, Baron Chanda, and members of the Chapman laboratory for their suggestions and comments regarding this manuscript. This study was supported by grants from the NIH (MH061876 and NS097362 to E.R.C.). H.B. was supported by a postdoctoral fellowship from the Human Frontier Science Program. E.R.C. is an Investigator of the Howard Hughes Medical Institute.

## Author contributions

D.D., H.B. and E.R.C. conceived of the project and designed the experiments. D.D., H.B., K.C.C. and L.W. performed the experiments. E.R.C. supervised the projects. D.D., H.B. and E.R.C. wrote the manuscript.

## Competing interests

The authors declare no competing interests.
