## [Peer Review File · Nature Communications]

Reviewers' Comments:

Reviewer #1:

Remarks to the Author:

This is an interesting paper. Chapman and coworkers previously developed a new assay to detect dynamics of single SNARE-mediated fusion pores between black membranes and nanodiscs. Through current measurements, the assay offers unprecedented resolution of fusion pore dynamics under well-controlled experimental conditions, including a fixed number of SNARE complexes involved in the fusion reaction and lipid composition. In this manuscript, Das et al. extended the approach to investigate the effect of synaptotagmin 1 (Syt) on fusion pore dynamics. They found that Syt blocks SNARE-mediated fusion in the absence of calcium, but promotes fusion in the presence of calcium. They also probed conformations of trans-SNAREs corresponding to different fusion pore states using single-molecule fluorescence and binding of soluble VAMP2 peptides. They observed that trans-SNARE complexes transit among many different conformations. Surprisingly, the assembled trans-SNARE complexes in the presence of Syt and calcium could be disassembled by NSF and alpha-SNAP, leading to closure of otherwise stable fusion pores. The manuscript is well-written, and experiments are carefully designed and performed. The experimental results are beautiful and generally support the main conclusions. Therefore, this manuscript has my enthusiastic recommendation for publication in Nature Communications, after revision to address my following comments:

1. The dynamic assembly and disassembly of a single SNARE complex have been widely studied by single-molecule force spectroscopy (Gao, Y. et al, Science, 2012; Zorman, S. et al, elife, 2014; Ma, L. et al, elife, 2015), under a condition that partially mimics the trans-SNARE complex. The authors are encouraged to discuss their findings on SNARE conformations with respect to those previous findings. In principle, SNARE mutations identified in those studies that specifically interfere with different SNARE folding/assembly steps can be used to dissect the SNARE conformations in the current assay. For example, since the committed trans-SNARE complexes are resistant to soluble VAMP2 binding, they are likely folded in the four-helical bundle region. Finally, it would be good to draw diagrams to illustrate different SNARE fusion states and fusion pore states (like Fig. 6c).
2. On Page 4, it says "Due to the larger diameter of ND1, membrane strain during pore formation would be lower than for ND3S, yet pores failed to form." It's unclear to me why the membrane strain is lower. Without experimental evidence, this sentence may be deleted.
3. Did any cis-SNARE complex form when ND9 was used? If not, what prevented its formation?
4. In figure 4, it would be better to show a time-dependent FRET trace to indicate transitions among different SNARE states.
5. It is widely believed that Munc18-1 protects trans-SNARE complexes from pre-mature disassembly by NSF. For a balanced discussion, the authors may point out this possibility, after the argument on the biological significance of NSF-dependent pore flickering on Pages 14-15.

Reviewer #2:

Remarks to the Author:

Review on the manuscript entitled "Resolving kinetic intermediates during the regulated assembly and disassembly of fusion pores" by Das, Bao and Chapman

The authors are using the setup they recently designed to study the effect of accessory proteins on the kinetics of SNARE-induced fusion pore opening and closing. Nanodiscs containing v-SNAREs are placed near a painted membrane (BLM) decorated with t-SNAREs. Current measurements provide a fast and accurate recording of pore kinetics. To obtain better information on the degree of SNARE assembly, they also performed single-molecule FRET experiments in the context of two apposing nanodiscs. They found that Synaptotagmin 1 (syt) favors the formation of a more zippered SNARE complex and stabilizes larger fusion pores that can only be closed by adding NSF. These results are interesting and worth publishing. However, their interpretation would require

some clarification before publication.

Major comments

1. Steric hindrance

Syt is a large protein containing notably 2 C2 calcium binding domains, each of them being globular with a ~4nm diameter. Fig 1a and Fig 6c are misleading because not at all to scale. They should be drawn to scale. This will show that the nanodiscs are very crowded (notably the small one) and suggest the presence of steric hindrance. The authors absolutely need to discuss this hindrance and how it may affect fusion.

2. Pore type and topology

a. Nowhere do the authors explicitly say that they assume the pore is proteinaceous. This is a plausible assumption that they already implicitly made in their previous article (Nature, 554, 260, e.g. Figs. 2a and 4d). However, this is not the common wisdom that assumes a fully connected lipid bilayer in which transmembrane domains can freely diffuse. This should be discussed and possibly tested by the standard bulk assay: lipid mixing and content release. If, in their hands, there is content release without lipid mixing, then a proteinaceous pore is very likely. On the other hand, if there is lipid mixing, there is no reason why free transmembrane domain would not zipper. It has been known for a decade (Nature, 460, 525 (2009)) that transmembrane domains of v-SNARE and t-SNARE can zipper. If they zipper, how can the same pore reopen? If they do not zipper, some interaction with another partner must be holding them apart. Could it be an interaction between the transmembrane domains and the MSP/NW proteins. Did the authors check this possibility?

b. How can it be shown that series of pore opening correspond to the same nanodisc and not to different ones? Is there a difference between the first opening and the subsequent ones?

c. Can a pore reseal by hemifusion instead of full fusion?

d. The size of the pore (Fig. 1d) deduced from the current (l. 129-138) needs to be better justified. The current roughly varies as r^2/l where r is the radius of the pore and l the thickness of the pore. An increase in current could be the consequence of a decrease in thickness without any change of radius (or, a change in both r and l). In the case of a proteinaceous pore, can a tilt of the transmembrane domain be consistent with a current increase by locally decreasing the thickness of the pore? i.e. a structure like: $|||><|||$ or $|||<>|||$, where $|$ represents the two bilayers and $<$ or $>$ the two apposing transmembrane domains. In any case, a putative (to scale) arrangement of the various proteins for each pore would greatly enhance the impact of this work.

3. Nanodisc-nanodisc vs. nanodisc-BLM: The single molecule FRET brings relevant information on the zippering state of the SNARE complex in the nanodisc-nanodisc setup. However, the geometry is not exactly the same as in the nanodisc-BLM system where the proteins, notably the t-SNAREs, are less constrained. The authors should briefly explain why they believe the results with the nanodisc-nanodisc setup are valid in the nanodisc-BLM system.

Minor comments:

Line 102: the argument about the density is not convincing. At these densities, the collision rates of the SNAREs are not significantly different. Hence their assembly rate should be similar. An alternate plausible explanation is that the transmembrane domains interact with the MSP and NW proteins, making the distance between the anchoring point too large to open a fusion pore (see 2a above). Alternate explanations may be suggested by the authors.

Line 162. Cannot a single exponential be explained by the fact that all SNARE complexes are placed at the same "primed" state by syt, making the pore opening the result of a single collective stroke? On the other hand, without syt, the SNARE complexes are zippered up to various levels

which could explain successive strokes.

Line 191: surprisingly there is still fusion without PIP2. Is it because of PS?

Line 219 and Fig. 4: Talking about C terminal is exaggerated. Labelling is at layer +4. The C terminal (up to layer +8 and the linker domain) may actually not be zippered.

Line 406: reguated \diamond regulated

Line 471: when is DPhPC used?

Line 507: The t-SNARE proteo-liposomes do not have the same lipid composition as the BLM. How does it modify the final lipid composition of the target membrane? As a corollary, what is the final concentration of t-SNAREs?

Line 518: MSP syb2 ratio = 2:1 for ND3. Is it correct? It was indicated 2:2 in the previous paper. ND7 was 2:10 in previous paper and ND9 here is 2:8 (but here it is a different scaffolding protein which may explain the difference).

Line 552: does not BAPTA bind 2 calcium ions? Then 1mM would bind to 2mM Ca^{2+} . Why are there 500 μM free?

Line 608: It corresponds to hydrophobic layers -7 and +4. This should be indicated (see comment about Line 219).

Frederic Pincet

Reviewer #3:

Remarks to the Author:

Disassembly of SNARE complexes after fusion and transmitter release is an active process that involves the ATPase NSF. It is well established that NSF disassembles cis SNARE complexes that are located in one membrane. In this manuscript the authors address the question if NSF also mediates the disassembly of trans SNARE complexes where the vSNARE and tSNAREs are anchored in two distinct membranes, the vesicle and plasma membrane respectively. Such a mechanisms could be involved in fusion pore closure without full fusion and thus could be significant for transient kiss-and-run fusion. The topic of this study is thus highly significant.

The authors address this question by measuring fusion pore currents with high time resolution in a nanodisc -black lipid membrane (ND-BLM) system. And also show some single molecule FRET (smFRET) measurements.

The authors show that synaptotagmin 1 (syt1) clamps spontaneous SNARE-induced fusion pore formation in the absence of Ca^{2+} but promotes more stable (increased lifetime), more dilated (with larger conductance) fusion pores in the presence of Ca^{2+} . This function of syt1 depends on the presence of PIP2. The smFRET experiments address the zippering states but the relation to fusion pore kinetics should be explained more clearly.

In Fig 6 the authors show that NSF in the presence of αSNAP and ATP induces closure of otherwise stable fusion pores.

This is an interesting and important study but several points should be addressed:

1. The main question is what is the evidence that the NSF acts actually on trans SNARE complexes. The fusion pores, which the authors measure in the presence of syt1 and Ca²⁺ have apparently a lifetime of 45 min. The authors assume that the SNAREs are still in a trans state but there is no direct evidence for this. This point should be discussed. Given the authors ability to perform smFRET experiments it might be possible to address this question using FRET labels at the syb2/syntaxin transmembrane domain C termini. Nevertheless, the observation that NSF induces fusion pore closure is a highly significant result.
2. p. 5: The authors show that v-SNARE density and not just copy number, is a crucial parameter for pore formation. This finding should be related to the results of Sharma and Lindau (2018) providing a mechanistic explanation for the requirement of SNARE complex proximity for fusion pore formation.
3. The description/terminology of fusion pore kinetics is not quite clear. The authors use the term of exponential kinetics for pore closure and pore opening but it seems that they actually refer to open and closed time distributions. This is confusing because the open time distribution actually reflects the kinetics of pore closure and vice versa. This should be clarified and is also relevant for the interpretation of the results.
4. Some recordings show multiple fusion pore openings separated by a closure with long duration, as in Fig. 3, lower trace. Is this the result of the re-opening of a pore formed by the same ND or a new pore formed by a different ND? This question relates to the analysis of pore closure dwell times and fraction of open times.
5. Related to this question is the pore lifetime in the presence of PIP₂. Do all pores eventually close or are they in an indefinite (on experimental time scale) stable open state? This is referred to in Fig 5 for the addition of cd-syb2 and BAPTA but in the preceding experiments it is not clear because open pore lifetimes are presumably limited by brief closing flickers. Is there some "terminal closure" indicated by long lifetime closed state and if so what is the total lifetime of the (somewhat flickering) fusion pores.
6. Fig. 4 and related text: How were the different FRET efficiency states quantified?
7. The time scale of the smFRET experiments is not fully clear. The authors state an initial 30 min incubation but what was the time scale following Ca²⁺ addition? How does the time scale relate to the waiting time until a fusion pore forms and the pore lifetimes?

Reviewer #4:

Remarks to the Author:

This manuscript utilized previously established ND-BLM system [Bao et al, 2018, Nature] together with ND-based smFRET to depict that Syt1, the major calcium sensor for synaptic exocytosis, promotes committed trans-SNARE complex formation thus maintaining and dilating fusion pore along with calcium. Further, the authors also found that the committed trans-SNARE complex can be disassembled by NSF/SNAP/ATP therefore driving fusion pore closure. In my view, to date, the ND-BLM system might be one of the most perfect platform for analyzing fusion pore kinetics in vitro due to its high sensitivity and time resolution. In addition, several phenomena are exciting as well. For instance, the critical role of PIP₂ during fusion pore opening and the clamping role of apo-Syt1 are reproduced in this system. In all, this is an interesting paper and I am in favor of publishing it if the authors can address my following questions.

Major comments:

1. ND9L (0.013 Syb2/nm²) vs. ND3s (0.022 Syb2/nm²): in the absence of syt1/Ca²⁺, why the former yields open pores whereas the latter yields closed pores regarding their similar Syb2

density (Fig. 1b and d).

2. Based on Fig. 1e, the author concluded that "inclusion of syt1 in both ND3s and ND9L reduced the occurrence of pores in the absence of Ca²⁺", (lines 116-117), however, in the trace of Fig. 1b, it seems to me that inclusion of syt1 increase the occurrence of pores in the absence of Ca²⁺. Please clarify this issue.

3. In lines 121-122, the conclusion "Hence, the clamping activity of apo-syt1 is strongly dependent on SNARE density and copy number" is confusing and overstated; if so, using higher density of apo-Syt1 while leaving SNARE densities unchanged, the clamping effect will be stronger. Thus, increasing apo-Syt1 copies/densities in ND3s or ND9L should be tested.

4. There is no strong evidence supporting the correlation between "syt1 dithers between the cis and trans" and "fusion pore flickering behavior" stated in lines 210-212. Otherwise, the author should analyze the effect of Syt1 copies/densities in the absence/presence of PI(4,5)P₂. More Syt1 is expected to efficiently bind limited SNAREs/SNARE complex thus quenching the flickering, which is important for the correlation.

5. Please specify the ND size used in Fig.4, since different NDs (ND3S, ND3L, or ND9L) may make a difference in CC-FRET efficiency.

6. In line 235, "In short, the ability of syt1 to clamp membrane fusion prior to the Ca²⁺ signal is associated with its ability to drive assembly of trans-SNARE complexes into a more zippered, yet inhibited, state....". However, the data in Fig. 4 only supports the more zippered model but not the inhibited model. Background and previous results on the clamped/inhibited role of syt1 in the absence of Ca²⁺ (for example, ref. 26 and other refs) should be discussed.

7. ND-based smFRET should be used to detect the role of NSF/SNAP/ATP in trans-SNARE complex disassembly.

8. Accessory proteins may preclude trans-SNARE complex disassembly by NSF/SNAP/ATP in order to protect the fusion pore formation during exocytosis, for example, HOPS in vacuolar fusion [Song et al, 2017, eLife]; and Munc18/Munc13 in synaptic exocytosis [Ma et al, 2013, Science]. The authors are encouraged to discuss these.

Minor comments:

1. In line 430, the authors should give a specific hypothesis testing method used in Fig.3c.

2. Chi-square tests were used throughout the manuscript, is it Pearson chi-square test? The author should precisely specify the exact statistical method.

3. In supplementary Fig.1c, the authors used "Kruskal-Wallis U-test", please check the words, is it "Kruskal-Wallis H-test"? or "Mann-Whitney U-test"?

4. In lines 616-617, the authors are suggested to rewrite the equation which calculates the FRET efficiency using equation editor.

Point-by-point responses to the reviewers' comments:

Reviewer #1 (Remarks to the Author):

The manuscript is well-written, and experiments are carefully designed and performed. The experimental results are beautiful and generally support the main conclusions. Therefore, this manuscript has my enthusiastic recommendation for publication in Nature Communications, after revision to address my following comments:

>> We very much appreciate the positive feedback from the referee.

1. The dynamic assembly and disassembly of a single SNARE complex have been widely studied by single-molecule force spectroscopy (Gao, Y. et al, Science, 2012; Zorman, S. et al, elife, 2014; Ma, L. et al, elife, 2015), under a condition that partially mimics the trans-SNARE complex. The authors are encouraged to discuss their findings on SNARE conformations with respect to those previous findings.

>> The reviewer makes an excellent point; we apologize for failing to cite the indicated single molecule studies from Yongli Zhang's laboratory (Gao, Y. et al. (2012) PMID: 22903523; Zorman, S. et al. (2014) PMID: 25180101; Ma, L. et al. (2015) PMID: 26701912). We now include these important references in our revised manuscript. Moreover, the revised manuscript now also includes a new figure (Supplementary Fig. 9a, also show immediately below) that illustrate the findings from the Zhang lab, along with the findings from our smFRET studies, in the context of SNARE complex structure. In the revised manuscript, we interpret our smFRET results in the context of the force measurements, stating: "We note that our smFRET experiments are in agreement with previous studies, based on force measurements, that examined the assembly of *trans*-SNARE complexes⁵²⁻⁵⁴ (Supplementary Fig. 9a). The emerging view is that SNARE assembly occurs in discrete steps that involve progressive zippering from their N- to C-termini, and this process is controlled by regulatory proteins."

Supplementary Fig. 9a: Structure of the SNARE motif four-helix bundle¹ showing the positions of the smFRET labels used in Fig.4; cy3 and cy5 are indicated with blue and red filled circles, respectively. Layers within the SNARE motifs are indicated by numbers and are shown within the SNARE complex as black polygons. The SNARE zippering states that were identified using force measurements²⁻⁴ are shown in magenta; where these coincide with layers in the SNARE complex, the polygon is rendered with alternating black and magenta stripes.

In principle, SNARE mutations identified in those studies that specifically interfere with different SNARE folding/assembly steps can be used to dissect the SNARE conformations in the current assay. For example, since the committed trans-SNARE complexes are resistant to soluble VAMP2 binding, they are likely folded in the four-helical bundle region.

>> This is again an excellent point (to study these mutant SNAREs in our ND-BLM assay). In fact, we had already planned a follow up paper precisely on this topic, but feel these experiments are beyond the purview of the current study; they will be the focus of our next study.

Finally, it would be good to draw diagrams to illustrate different SNARE fusion states and fusion pore states (like Fig. 6c).

>> To address this issue, we now include four new illustrations. In two (revised versions of Fig. 1a and Fig. 6c), we provide scale drawings of all components for each type of ND used in our study. Then, in a new figure (Supplementary Fig. 9a, discussed above), we illustrate the SNARE zippering states using the crystal structure of the SNARE complex as a template; we have annotated this with our smFRET findings as well as the findings from the force measurements from the Zhang lab, again as detailed above. Finally, we added another new figure (Supplementary Fig. 12), in which we provide scale drawings of all the components, and indicate the size estimates for the fusion pore under each condition in our study. This new figure is shown below:

Supplementary Fig. 12

Supplementary Fig. 12: Scale drawing of the fusion pore components used in the current study. Pore diameter estimates, obtained from the conductance values shown in Supplementary Fig. 3, are shown for each condition. The black and purple lines indicate the TMDs of syntaxin 1A, syb2, respectively; the green lines indicated the linkers and luminal domain of syt1.

2. On Page 4, it says “Due to the larger diameter of NDL, membrane strain during pore formation would be lower than for ND3S, yet pores failed to form.” It’s unclear to me why the membrane strain is lower. Without experimental evidence, this sentence may be deleted.

>> As suggested by the referee, we have deleted this sentence.

3. Did any *cis*-SNARE complex form when ND9 was used? If not, what prevented its formation?

>> Pores formed by ND9, without synaptotagmin 1 (syt1), were closed by the addition of the cytoplasmic domain of synaptobrevin 2 (cd-syb2) (Figure 5a, upper panel). This result reveals that fully zippered *cis*-SNARE complexes did not mediate the formation of these pores because the off-rate of SNAREs in fully zippered *cis*-SNARE complexes is virtually zero. So, there would be no binding site for cd-syb2 to join into a SNARE complex to close a fusion pore; we conclude the SNARE complex must be in a *trans* configuration under these conditions.

Pores that were opened by the action of Ca^{2+} •syt1 were not closed by cd-syb. However, these pores were closed by the action of NSF. Since there is no possible physical mechanism by which the disassembly of fully zippered *cis*-SNARE complexes (which are defined as having all three SNAREs fully zippered together all the way through the transmembrane domains (TMD) **in the same membrane**) can close a pore, we conclude that these pores were also formed by *trans*- and not *cis*-, SNARE complexes. That is, if the v- and t-SNAREs TMDs came together, we see no process by which NSF-mediated disassembly could, in effect 'segregate' the two bilayers, to close pores. SNAREs do work on bilayers, to form pores and fuse membranes, by pulling bilayers together, but their disassembly - by NSF - does not 'reciprocal work' to separate the bilayers and thus close a pore.

We note that cd-syb2 did not close 100% of pores (formed in the absence of Ca^{2+} •syt1), and the resistant fraction could represent a small population of fully zippered *cis*-SNARE complexes. The same is true for the NSF experiments conducted on pores formed by the action of Ca^{2+} •syt1: in 1 out of 7 trials NSF failed to close a pore, and this lone trial could reflect fully zippered SNARE complexes.

Regarding what prevents formation of fully zippered (through the TMDs) *cis*-SNARE complexes, as raised by the other reviewers it is possible that the MSP interacts with the syb2 TMD, and prevents it from binding to the TMD of syntaxin. This would be fortuitous, as our goal has always been to find any way to trap pore intermediates so that we can study them. We have begun to test this possibility using a pull-down assay, as shown below:

Supplementary Fig. 13 – Immobilized MSP13 and MSP30 fail to pull-down syb2. Pull-down assays were carried out by first incubating Ni-beads with his6-MSP13 or his6-MSP30; free MSP was removed by washing 3x with 50 mM imidazole. Purified full-length syb2 was then incubated with bead-bound MSP in detergent (1% OG) at 4°C. Beads were washed three times in binding buffer (25 mM HEPES (pH: 7.5), 100 mM KCl and 1% OG) containing 50 mM imidazole.

Three samples from each assay were subjected to SDS-PAGE: total (T; input), beads (B; bead-bound material after washing), and supernatant (S; unbound material that was removed before washing). In each case, 10% of the samples were loaded, and proteins were visualized by staining with Coomassie blue. No apparent interaction between syb2 and either MSP was apparent under these conditions.

In these experiments, we did not observe interactions between MSP and syb2. Of course, this result must be interpreted with caution, as the pull-down had to be conducted in the presence of detergent. It remains possible (and is perhaps even likely) that in lipid-filled NDs, the syb2 TMD interacts with the MSP. In our revision, we provide the pull-down results in Supplementary Fig. 13, and state: “However, since these experiments were performed in presence of detergent, it remains formally possible that interactions might occur within lipid-filled NDs”. We continue to explore this issue, and will focus on a follow-up paper that delves into the mechanism by which SNAREs are arrested in a *trans* configuration in intact NDs as follows: cross-linking the MSP to the TMDs (which is challenging, as they are in such close proximity that some degree of cross-linking will surely occur even if there is no interaction), FRET between the MSP and TMDs (again, challenging because of the close proximity), and ensemble FRET between the C-termini of v- and t-SNAREs. We have conducted these ensemble FRET experiments (using the same donor and acceptor used in the smFRET experiments, and using samples that contained Ca^{2+} •syt1 to maximize SNARE complex assembly) to address this latter issue. In these ensemble experiments (which are described in additional detail in our response to reviewer #3 point #1), we observed clear donor quenching (18 + 5%) when the SNARE motifs were labeled (again, using the same dyes and labeling positions as in the smFRET experiments). In contrast, when we examine FRET using the same dyes placed on a cysteine residue at the C-terminal ends of both syb2 and syntaxin 1, in some trials we saw no effect on donor quenching, which would provide direct evidence that the v- and t-SNARE TMDs fail to coalesce in our system. However, in some trials, we observed an anomalous increase in the emission of the donor, which confounded our interpretation. We are working to resolve this issue, and these experiments will include entirely new work that examines how the local environment of labels on the C-termini are affected as SNARE pairing occurs. Our working model is that the C-termini might be pulled up toward, or into, the bilayer, during fusion, thus impacting the fluorescence signals. We feel this new fluorescence work is beyond the purview of the current manuscript; we will follow-up on this matter in a subsequent study. Regardless, we argue that this does not impact the findings in our current study, as the cd-syb2 and NSF experiments indicate we are studying *trans*, and not *cis*, complexes.

Finally, we should note that it is also formally possible that the high degree of curvature prevents SNARE TMDs from moving toward one-another, because nothing is known about how the TMDs diffuse within a highly curved membrane. Also, while we have evidence that nascent fusion pores formed using NDs are hybrid structures, composed of both lipids and the TMDs of SNARE proteins (Bao et al. (2016) PMID: 26656855), the degree to which the nascent pore is lipidic or proteinaceous remains unclear. If the nascent pore is largely proteinaceous, a diffusion path for the TMDs to coalesce with one-another might not be available. We briefly make this points in the revised manuscript.

4. In figure 4, it would be better to show a time-dependent FRET trace to indicate transitions among different SNARE states.

>> To address this issue, we now include representative real-time traces of low, medium and high FRET states in Supplementary Fig. 9b; this figure is also provided below:

Supplementary Fig. 9b. Four representative traces, obtained using the CC FRET pair in the presence of *syt1* and Ca^{2+} , are shown. Under this condition, all three FRET states were observed, corresponding to low (0.2), medium transition to low (0.6 to 0.2), high transition to medium (0.8 to 0.6), and high (0.8) states.

Revised manuscript now states: “For completeness, representative raw time-based FRET traces are shown in Supplementary Fig. 9b, for samples that contained Ca^{2+} and *syt1*; all FRET states are illustrated in these samples.”

5. It is widely believed that Munc18-1 protects trans-SNARE complexes from pre-mature disassembly by NSF. For a balanced discussion, the authors may point out this possibility, after the argument on the biological significance of NSF-dependent pore flickering on Pages 14-15.

>> We thank the referee for this suggestion, and have included this point in the revised Discussion, which states: “Interestingly, binding of SM (Sec1-Munc18) proteins in synapses⁵⁸ or the binding of HOPS (homotypic fusion and protein sorting) complex in case of vacuolar fusion⁵⁹ might protect *trans*-SNARE complexes from NSF/ α -SNAP-mediated disassembly⁶⁰. It will be interesting to determine whether these factors affect NSF mediated fusion pore closure in the ND-BLM system described here.”

Reviewer #2 (Remarks to the Author):

These results are interesting and worth publishing.

>> We are pleased that the referee found our work to be interesting.

Major comments

1. Steric hindrance

Syt is a large protein containing notably 2 C2 calcium binding domains, each of them being globular with a ~4nm diameter. Fig 1a and Fig 6c are misleading because not at all to scale. They should be drawn to scale.

>> As suggested, we have re-drawn these two figures (Fig. 1a and 6c) to scale in the revised manuscript. Please also see our response to referee #1, point #1 (3rd part), where we also provide new figures to further address this issue (Supplementary Figs. 9a and 12).

This will show that the nanodiscs are very crowded (notably the small one) and suggest the presence of steric hindrance. The authors absolutely need to discuss this hindrance and how it may affect fusion.

>> This is an interesting, and somewhat complicated point. First, we note that the physiological density of syb2 on synaptic vesicles is 0.014 copies/nm² (assuming 70 copies per 40 nm synaptic vesicle from Takamori et al. (2006) PMID: 17110340). For syt1 (assuming 15-20 copies, again from Takamori et al. (2006) PMID: 17110340), the density is 0.003-0.004 copies/nm². We calculated our syb and syt1 densities (note: in all experiments - except for a new figure where we titrated-down syt1 - syb and syt were reconstituted 1:1), as follows:

$$\begin{aligned} \text{ND3}_s &= 0.02 \text{ (0.01) copies/nm}^2 \\ \text{ND3}_L &= 0.004 \text{ (0.002) copies/nm}^2 \\ \text{ND9}_L &= 0.013 \text{ (0.06) copies nm}^2 \end{aligned}$$

Because the reconstitution is random, only half of these proteins are on a given side of the nanodiscs; these values are shown in brackets, above. So, the syb2 densities are at or below physiological densities, and syt1 densities range from above, to below, physiological densities (please see new Supplementary Fig. 2, where we titrated the copy number of syt1 down from 9, 5 and 2 copies per ND_L). Moreover, synaptic vesicles contain a myriad of other proteins, so we argue that they are much more crowded than our nanodiscs. Regardless, the degree of crowding is a fair point, so as suggested, we now illustrate how crowded the NDs are in a scale models shown in Fig. 1a and 6c, and we comment on the fact the NDs are crowded in the revised text.

Perhaps the reviewer is concerned that steric hindrance by syt1 serves to clamp fusion, by preventing NDs from docking onto the BLM. This idea is complicated by reports that syt1 helps to dock together the two membranes that are destined to fuse (e.g. Reist et al. (1998) PMID: 9742137, Wang et al. (2011) PMID: 22184197; among many other examples); that is, syt1 appears to promote docking, not to interfere with docking. Also, we previously showed that the conversion of syt1 from a fusion clamp to an activator of fusion involves a conformational change that alters the relative disposition of the tandem C2-domains (Bai et al. (2016) PMID: 27001899). The mass on the surface of the ND does not change, just the angle between the tandem C2-domains. So, we argue that syt1 does not simply ‘move out of the way’ to allow fusion to proceed.

Finally, perhaps the referee is alluding to the idea that adding mass can potentially drive pore dilation via steric effects between SNAREs. Ca²⁺ and syt1 might drive the formation of larger pores via such a steric mechanism. We are currently planning future follow-up studies to test the idea that adding (benign) mass might help to drive pore dilation (by steric effects).

We reiterate that during revision we conducted new experiments in which we titrated-down the syt1 copy number (Supplementary Fig. 2). We found that syt1 inhibits pore activity, at least in part, by reducing the open life time in a dose-dependent manner; this effect is independent of pore occurrence, so aspects of the inhibitory/clamping activity of syt1 appear to be due to intrinsic properties of the protein.

In the manuscript we now state: “Syt1 is unlikely to clamp fusion by preventing the docking of NDs to the BLM via steric effects, as this protein has been reported to facilitate docking in reconstituted systems and in synapses^{36,51}. Moreover, we used physiologically relevant syb2

and syt1 densities, as compared to native synaptic vesicles³⁰. Indeed, synaptic vesicles contain a myriad of additional proteins, so are more crowded than our NDs. Finally, the observation that syt1 inhibits pore activity, at least in part, by reducing the open life time in a dose-dependent manner (Supplementary Fig. 2), indicates that aspects of the inhibitory/clamping activity of syt1 are due to intrinsic properties of the protein.”

2. Pore type and topology

a. Nowhere do the authors explicitly say that they assume the pore is proteinaceous. This is a plausible assumption that they already implicitly made in their previous article (Nature, 554, 260, e.g. Figs. 2a and 4d). However, this is not the common wisdom that assumes a fully connected lipid bilayer in which transmembrane domains can freely diffuse. This should be discussed and possibly tested by the standard bulk assay: lipid mixing and content release.

>> We apologize for not making this clear. To clarify, please note that this issue was addressed by our lab using NDs (bearing syb2 and syt1) and SUVs (bearing t-SNAREs) in Bao et al. (2016) PMID: 26656855; Supplementary Figure 4. We observed that content release was always associated with lipid mixing. From these, and other experiments reported in our 2016 study, we concluded that fusion pores are hybrid structures, composed of both the transmembrane domains (TMDs) of SNAREs and lipids. For example, we showed that the TMDs were exposed to solvent (and thus were exposed to the lumen of fusion pores), as a single face of the TMD helix was labeled with water soluble probes. However, pores can form with as few as two SNARE complexes; under this condition the pore cannot be purely proteinaceous. To address this concern, in revision, we are careful to explicitly state that fusion pores are hybrid structures.

Please also see our response to point #3 from referee #1, where we discuss our findings that the TMDs of v- and t-SNAREs rarely coalesce in our system.

If, in their hands, there is content release without lipid mixing, then a proteinaceous pore is very likely. On the other hand, if there is lipid mixing, there is no reason why free transmembrane domain would not zipper. It has been known for a decade (Nature, 460, 525 (2009)) that transmembrane domains of v-SNARE and t-SNARE can zipper. If they zipper, how can the same pore reopen? If they do not zipper, some interaction with another partner must be holding them apart.

>> In many of our experiments, we use ND3, which have an average of 1.5 copies of SNAREs per face of the ND (so, one or two copies). It was reported previously by the Pincet lab (Shi, et al. (2012) PMID: 22422984) and by Sinha et al. (2011) (PMID: 21844343) that ~2-3 SNAREs are enough to open a fusion pore. In principle, if we increase SNARE copy numbers, it might be possible to assemble purely proteinaceous pores. But, again, the results from our laboratory indicate that even at high SNARE copy numbers, the fusion pore is a hybrid structure of both proteins and lipids (Bao et al. (2016) PMID: 26656855); at very high protein densities we find the bilayers leak, so there are limitations as to how high we can go.

We argue the pores flicker open and closed, for an extended period of time, because the SNARE complexes fail to fully form, and the TMDs rarely zipper together into fully assembled *cis*-SNARE complexes. Our finding that the TMDs rarely come together in our system is discussed in detail above, in response to point #3 from reviewer #1 above, where we also speculate as to why the TMDs rarely, if ever, come together to form fully *cis*-SNARE complexes.

Could it be an interaction between the transmembrane domains and the MSP/NW proteins. Did the authors check this possibility?

>> Yes, this is certainly a possibility, and we have conducted experiments to address this issue – again, please see our response to point #3 from reviewer #1 above, where we discuss these experiments.

b. How can it be shown that series of pore opening correspond to the same nanodisc and not to different ones?

>> We have carefully titrated and standardized the concentrations of ND in our ND-BLM assay in order to obtain individual pores. That is, only low levels of NDs are used, low enough to ensure we usually obtain a single pore. Moreover, pores open and close in one step, either by themselves, or upon addition of cd-syb2 or NSF. This observation demonstrates these are individual pores; if there was more than one pore, all openings and closings would not occur at the exact same time. We do occasionally observe multiple pores, and such recordings are discarded; an example is provided below:

Figure legend: Multiple pore formation in the ND-BLM system. Figure shows a rare example of when multiple pore formation occurred in our current study, indicating the docking of multiple NDs onto the same BLM, with the opening of three distinct pores. Please note the subtle differences in the currents detected for each different pore; this is the result of heterogeneity in terms of SNARE copy number per ND during reconstitution, as documented in Bao et al. (2018) PMID: 29420480.

From the trace it is evident that the currents/conductance values of distinct, individual pores are slightly different. We addressed this issue of the heterogeneity of our ND preparations in our previous paper (Bao et al. (2018) PMID: 29420480 - Figure 3c). In short, when we reconstitute the v-SNAREs, there is always some variation in the copy number between individual NDs. For example, a preparation with a mean of 5 copies of syb2 (ND5) will also contain some NDs that have fewer than 5 copies, as well as NDs with more than 5 copies. As we showed in our 2018 study (Bao et al.), the pore current depends on the copy number, so an ND5 preparation will have currents from pores formed by NDs with fewer than, and more than, 5 copies. So, if a second pore, from a different ND, opens within in the same trace, we would expect to observe (in most cases) a somewhat distinct conductance value for the second pore. We did not observe this within an individual trace, so conclude our traces contain a single pore. Importantly, we carefully titrated the [ND] such that, in the vast majority of experiments, only a single pore forms; only rarely did we observe the formation of more than one pore, so the example of three pores, in the recording shown above, is a rare occurrence. Regardless, when multiple pores occur, those recordings are discarded; only trial that yielded a single pore were analyzed.

Is there a difference between the first opening and the subsequent ones?

>> No, there is no difference in the first opening and the subsequent ones. This is formally addressed as follows:

a) The current always stays the same (again demonstrating that we are looking at opening and closure of the same pore). A representative example with different epochs of the ND9_L + syt1 pore (in presence of Ca²⁺) is shown below along with the current histograms (note: for a representative example of a pore formed by SNAREs alone, please see Extended data Figure 4b of Bao et al. (2018) PMID: 29420480, where we document that the current and kinetic properties of pores do not change over the course of a recording). It is apparent from this analysis that the current through an open pore does not change over the course of a recording.

b) The kinetic behavior also stays the same at the beginning, middle and end of a trace. Please see the bottom panel in the figure below, where the open time distributions from three distinct epochs from the same trace are shown. Again, the kinetic properties of an individual pore do not change over the course of a recording.

Figure legend. Fusion pore properties do not differ at the beginning, middle, or end of a recording in the ND–BLM assay. Raw traces, and current and open time histograms of an individual ND9_L + syt1 pore in the presence of Ca²⁺, at different time points in the recording, are shown. There were no significant differences at the beginning, middle, or end of a recording session. The baseline was stable over the course of all recordings. Closed (C) and open (O) states are shown, pore current is indicated.

These are typical findings observed in all of our recordings. We now mention these findings in our revised manuscript, which states: “Pore properties did not differ at the beginning, middle, or end of a recording, but all pores closed within ~90 minutes.”

c. Can a pore reseal by hemifusion instead of full fusion?

>> We agree that reversion to a hemi-fused state could lead to pore re-sealing, as first proposed in a ground-breaking paper from the Pincet lab (Shi et al. (2012) PMID: 22422984); we feel this is a likely mechanism for closure in our experiments. We now state this in the revised manuscript: “Once the terminal closure occurred, there were no further openings or flickers. Closure might involve reversion to a hemi-fused state²⁴”.

d. The size of the pore (Fig. 1d) deduced from the current (l. 129-138) needs to be better justified. The current roughly varies as r^2/l where r is the radius of the pore and l the thickness of the pore. An increase in current could be the consequence of a decrease in thickness without any change of radius (or, a change in both r and l). In the case of a proteinaceous pore, can a tilt of the transmembrane domain be consistent with a current increase by locally decreasing the thickness of the pore? i.e. a structure like: $|||> <|||$ or $|||< >|||$, where $|$ represents the two bilayers and $<$ or $>$ the two apposing transmembrane domains. In any case, a putative (to scale) arrangement of the various proteins for each pore would greatly enhance the impact of this work.

>> We agree that the pore current is dependent on the radius as well as the length of the pore, and that there are uncertainties regarding pore length, so the pore diameter can only be estimated from conductance values. We now emphasize this point in the revised manuscript by stating: "Because the length dimension of a fusion pore is not known, and because this parameter might change with tilting of the SNARE TMDs, our pore diameter values based on conductance measurements are only approximations".

As suggested, we now include illustrations, drawn to scale, of the various proteins, nanodiscs, and fusion pores; this is detailed in our response to referee #1 point #1 (last section of our response).

3. Nanodisc-nanodisc vs. nanodisc-BLM: The single molecule FRET brings relevant information on the zippering state of the SNARE complex in the nanodisc-nanodisc setup. However, the geometry is not exactly the same as in the nanodisc-BLM system where the proteins, notably the t-SNAREs, are less constrained. The authors should briefly explain why they believe the results with the nanodisc-nanodisc setup are valid in the nanodisc-BLM system.

>> We comment on this caveat in the Discussion of the revised manuscript as follows: "It was not technically feasible to conduct smFRET using v-SNARE NDs bound to t-SNARE BLMs, so we used v- and t-SNARE NDs. The strength of this approach is that the assembly state of full-length, membrane embedded SNAREs can be assessed, but we note that the t-SNARE NDs are more constrained as compared to the BLM. Nonetheless, this system revealed that apo- and Ca^{2+} -bound syt1 have distinct, direct effects on the assembly of *trans*-SNARE complexes."

The general view is that SNAREs assemble from N-to-C in all systems, ranging from the force experiments based on cytoplasmic domains of SNAREs (detailed above, from Yongli Zhang's laboratory, and others), to our ND-ND smFRET experiments (Figure 4), to our ND-BLM system (e.g. in Bao et al. (2018) PMID: 29420480, a small truncation at the C-terminus of SNAP-25 allowed pores to form, but destabilized them, suggesting N-terminal zippering occurred, but impairment of C-terminal zippering destabilized the open state). So, in this respect, we argue that these systems are at least somewhat comparable.

Minor comments:

Line 102: the argument about the density is not convincing. At these densities, the collision rates of the SNAREs are not significantly different. Hence their assembly rate should be similar.

>> We note that it has been reported that collision is not rate limiting (Schuette et al, (2004) PMID: 14981239). Rather, docking appears to be rate limiting (Smith et al. (2011) PMID: 21539781). According to this view, multiple collisions occur, and the rate limiting step is whether v- and t-SNAREs manage to 'grab' one another, and this involves conformational changes in SNAREs (e.g. the SNARE motif of syb2 is largely unstructured, until it binds t-SNAREs; the

Habc domain of syntaxin impedes v-/t-SNARE interactions [Dulubova et al. (1999) PMID: 10449403, MacDonald et al. (2010) PMID: 20074061], and when this domain is deleted the rate of assembly, and thus fusion, is dramatically enhanced, among other examples). Indeed, if v- and t-SNARE vesicles are artificially ‘docked’ with one-another, fusion is greatly accelerated (Hui et al. (2011) PMID: 21642967). We argue that as we increase the v-SNARE copy number per ND, the chances that a given collision will include SNAREs that are in a ‘productive’ state/conformation is increased, thus enhancing the rate of assembly.

An alternate plausible explanation is that the transmembrane domains interact with the MSP and NW proteins, making the distance between the anchoring point too large to open a fusion pore (see 2a above). Alternate explanations may be suggested by the authors.

>> Please see our response to referee #1 point #3, where we discuss whether the scaffolding proteins bind syb2, and provide new data that help to address this issue.

Line 162. Cannot a single exponential be explained by the fact that all SNARE complexes are placed a the same “primed” state by syt, making the pore opening the result of a single collective stroke? On the other hand, without syt, the SNARE complexes are zippered up to various levels which could explain successive strokes.

>> We agree, completely; this was the point we tried to make, and so we apologize if this was unclear. To clarify further, we have changed the text in our revised manuscript to state: “In the absence of Ca^{2+} , spontaneous openings still occurred to some degree, and kinetic analysis revealed that these openings followed multi-exponential kinetics (Fig. 2c), indicating the involvement of multiple intermediates. In contrast, in the presence of both Ca^{2+} and syt1, pore opening followed single exponential kinetics, suggesting that all trans-SNARE complexes were driven into the same primed state, making the pore opening the result of a single collective stroke.”

Line 191: surprisingly there is still fusion without PIP₂. Is it because of PS?

>> Yes, fusion in the absence of PIP₂ is strictly dependent on the presence of PS. We were the first to omit PS in the standard *in vitro* lipid mixing assay (Bhalla et al. (2005) PMID: 16093350); we reported that omission of PS from both the v- and t- populations of liposomes completely abrogated the ability of three isoforms of syt (I, VII, and IX) to stimulate fusion.

We have also performed ND-BLM assays without PS (or PIP₂) in the BLM, results are provided below (Supplementary Fig. 8) where it is apparent that Ca^{2+} and syt1 are unable to regulate pores in the absence of PS. We now mention these findings in the revised manuscript: “As a further control, both PS and PIP₂ were omitted from the BLM; under this condition, Ca^{2+} and syt1 were unable to regulate pores (Supplementary Fig. 8). These experiments confirm that syt1-lipid interactions play a key role in the regulation of fusion pores.”

Supplementary Fig. 8

Supplementary Fig. 8. Ca²⁺•syt1 is unable to regulate fusion pore formation in the absence of PS in the BLM. Fraction of trials in which a fusion pore was detected, plotted as % occurrence, using ND_{3S}/syt1 and ND_{3L}/syt1 in the presence (+) or absence (-) of Ca²⁺, as a function of the indicated lipid composition. Three independent sets of NDs of each type were used for left and middle panel data sets, four independent sets were used for the right panel, and the total number of measurements obtained under each condition (n) is indicated. Pearson's χ^2 analysis was performed, *** $p < 0.001$.

Line 219 and Fig. 4: Talking about C terminal is exaggerated. Labelling is at layer +4. The C terminal (up to layer +8 and the linker domain) may actually not be zippered.

>> This is an excellent point, and it has been addressed in revision with a new illustration (Supplementary Fig. 9a), showing each layer of the SNARE complex, and indicating our smFRET results as well as the findings from the force measurements from the Zhang lab (as detailed in our response to point #1 from referee #1 above). We have also corrected the exaggerated statement in the text.

Line 406: reguated \diamond regulated

>> Corrected

Line 471: when is DPhPC used?

>> We used DPhPC in all our BLM lipid preparations. We clarified this in the text: "Planar lipid bilayer recordings were performed using a Planar Lipid Bilayer Workstation (BLM) from Warner Instruments (USA) as described^{16,63,64}. Briefly, lipids (30% DOPE, 52% DPhPC, 16% DOPS and 2% brain PIP₂, at 30 mg/ml in *n*-decane) were first painted onto a 150- μ m aperture in a 1 ml, white Delrin or polystyrene cup (Warner Instruments) and dried for 15 minutes."

Line 507: The t-SNARE proteo-liposomes do not have the same lipid composition as the BLM. How does it modify the final lipid composition of the target membrane?

>> In our experiments ~40 t-SNARE vesicles (of ~40 nm diameter) fuse with the BLM (for method of detection, please see Bao et al. (2018) PMID: 29420480, Supplementary Figure 3c). The hole onto which we paint the BLM is 150 μ m diameter. When t-SNARE vesicles fuse with the BLM, they change the lipid composition of the BLM by **0.0011%**. We clarified this in the text:

“In our experiments ~40 t-SNARE vesicles (of ~40 nm diameter) fuse with the BLM¹⁶. The hole at which BLM lipid is painted is 150 μm diameter. When t-SNARE vesicles fuse with the BLM, they change the lipid composition of the BLM by 0.0011%.”

As a corollary, what is the final concentration of t-SNAREs?

>> t-SNAREs were reconstituted into black lipid membranes (BLMs), at a density of 0.4 molecules per μm². This was determined in Supplementary Figure 3c, d in our previous paper on this topic (Bao et al. (2018) PMID: 29420480). We apologize for not making this clear in the original version of our study, and this is now stated in the revised Methods section: “t-SNAREs were reconstituted into black lipid membranes (BLMs), at a density of 0.4 molecules per μm²¹⁶.”

Line 518: MSP syb2 ratio = 2:1 for ND3. Is it correct? It was indicated 2:2 in the previous paper. ND7 was 2:10 in previous paper and ND9 here is 2:8 (but here it is a different scaffolding protein which may explain the difference).

>> To provide a clear answer, we broke this question down into three parts:

a) *“MSP syb2 ratio = 2:1 for ND3. Is it correct?”*

>> Yes, the MSP syb2 ratio used to reconstitute ND3 was 2:1. The methods section of our manuscript now states: “...the MSP to syb2 ratios were 2:1 (ND3) and 2:8 (ND9).”

b) *“It was indicated 2:2 in the previous paper.”*

We apologize for this confusion. To clarify, the 2018 *Nature* paper (PMID: 29420480) stated: “Reconstitution of SYB2 into 13-nm nanodiscs was performed as described⁹. For reconstitution of SYB2 into 50-nm nanodiscs, the MSP:lipid ratio was 2:4,000. To incorporate different copy numbers of SYB2 into **50-nm nanodiscs**, the following MSP:SYB2 ratios were used: **2:2 (ND3)**, 2:4 (ND5) and 2:10 (ND7).” **So, the 2:2 ratio was only for the 50 nm NDs used in the supplement of that study.** Moreover, reference 9 in the 2018 paper was our *NSMB* paper from 2016 (PMID: 26656855), which stated: “To prepare nanodiscs containing different copy numbers of syb2, the following MSP/syb2 ratios were used: 2:0.2 (ND1), 2:0.4 (ND2), **2:1 (ND3)**, 2:2 (ND4), 2:4 (ND5), 2:6 (ND6), 2:8 (ND7) and 2:10 (ND8).” So, we referenced a study in which the ratio was 2:1.

c) *“ND7 was 2:10 in previous paper and ND9 here is 2:8 (but here it is a different scaffolding protein which may explain the difference).”*

>> Yes, the reason is that we used a different scaffolding protein. For example, the ND7 2:10 ratio was used for the 50 nm NDs in our 2018 paper, and the ND9 2:8 ratio was for 30 nm NDs used in the current study.

Line 552: does not BAPTA bind 2 calcium ions? Then 1mM would bind to 2mM Ca²⁺. Why are there 500μM free?

>> We respectfully disagree; below, we provide references in which number of calcium ions binding to BAPTA was quantified. From these studies, 1 BAPTA binds to 1 Ca²⁺ ion. There is sometimes confusion on this point (e.g. the Wiki entry is incorrect). Part of this might be due to the fact that two protons are lost each time one Ca²⁺ ion binds.

1. Tsien, R.Y. *Biochemistry*. 1980,19(11):2396-404. (First paper to report synthesis of BAPTA).
2. Bootman, M.D. et al. *Cell Calcium*. 2018, 73:82-87.
3. Greig et al. *J Inorganic Biochemistry*. 1987, 113-121.
4. Gillies, R.J. *NMR In Physiology and Biomedicine*, page 265.

Given the confusion about this issue, we used a Ca^{2+} indicator Indo 1 and tested the ability of BAPTA to chelate Ca^{2+} . These results are now provided as Supplementary Fig. 1c, and are also shown below:

Supplementary Fig. 1c: BAPTA- Ca^{2+} stoichiometry determination by Indo 1 (10 μM) fluorescence; excitation was at 340 nm. The [BAPTA] and [Ca^{2+}] under each condition are indicated.

We observed that binding of Ca^{2+} causes a blue-shifted increase in Indo-1 fluorescence, as expected (Grynkiewicz, G. et al. (1985) PMID: 3838314 - the first paper to describe Indo-1). In principle, if BAPTA bound two molecules of Ca^{2+} , then 5 μM BAPTA would chelate 10 μM Ca^{2+} and prevent a change in Indo-1 fluorescence. Notably, we found that the Indo-1 fluorescence spectra are comparable between the 0 mM BAPTA/5 μM Ca^{2+} condition (blue curve) and the 5 μM BAPTA/10 μM Ca^{2+} condition (dark red curve). This validates that one molecule of BAPTA binds one molecule of Ca^{2+} .

Our revised manuscript now states: “When syt1 alone was reconstituted into NDs (ND0), pores failed to form either in presence or absence of Ca^{2+} (Supplementary Fig. 1b), confirming that pore formation required *trans*-SNARE pairing. For the Ca^{2+} -free conditions, BAPTA was used to chelate any residual Ca^{2+} present in the buffers. In most of our experiments, we subsequently added Ca^{2+} to yield the indicated [Ca^{2+}]_{free}, so we confirmed that BAPTA does indeed bind Ca^{2+} with a stoichiometry of 1:1 (Supplementary Fig. 1c).

Line 608: It corresponds to hydrophobic layers -7 and +4. This should be indicated (see comment about Line 219).

>> Again, this is an excellent point, and as indicated above in response to referee #1, point #1, this issue has been addressed in revision with a new illustration (Supplementary Fig. 9a), showing each layer of the SNARE complex, and indicating our smFRET results as well as the force measurement results from the Zhang lab.

Frederic Pincet

Reviewer #3 (Remarks to the Author):

The topic of this study is thus highly significant.

This is an interesting and important study but several points should be addressed:

>> We are gratified that the referee found our study to be significant, interesting and important.

1. The main question is what is the evidence that the NSF acts actually on trans SNARE complexes. The fusion pores, which the authors measure in the presence of syt1 and Ca²⁺ have apparently a lifetime of 45 min. The authors assume that the SNAREs are still in a trans state but there is no direct evidence for this. This point should be discussed. Given the authors ability to perform smFRET experiments it might be possible to address this question using FRET labels at the syb2/syntaxin transmembrane domain C termini. Nevertheless, the observation that NSF induces fusion pore closure is a highly significant result.

>> We considered smFRET experiments using labels at the C-termini of the SNARE transmembrane domains. However, more than two *trans*-SNARE complexes are required to open a fusion pore in the ND-BLM system. Monitoring multiple *trans*-SNARE complexes by smFRET is extremely challenging, and we are in the process of developing new single molecule approaches.

Instead, we conducted ensemble FRET experiments; please see our response to reviewer #1, point #3 above, for a detailed description of this issue and our findings. Briefly, for these experiments, we used NDs bearing syb2 and syt1, and liposomes that harbored t-SNARE heterodimers (to better mimic the BLM), in the presence of Ca²⁺ (to drive SNAREs into the committed state). We observed that after a 60 min incubation, FRET - as evidenced by donor quenching - was observed when the dyes were placed in the SNARE motifs (in the same position as in the smFRET experiments). In contrast, FRET was not observed when the dyes were placed at the C-termini of syb2 and syntaxin. However, in some of the C-terminal FRET trials, we observed an anomalous increase in the emission of the donor. Our working model is that the local environment of the dye on the C-terminus might change during SNARE pairing, potentially pulling the dye against, or even into, the bilayer. We continue to conduct experiments to address the variation in the C-terminal FRET measurements, but feel that this complicated issue is beyond the scope of the current manuscript; it will be the focus of a follow-up study. Finally, we emphasize that the cd-syb and NSF experiments – while indirect – strongly indicate that we are studying *trans*, and not *cis*, complexes, as neither treatment would be able to close a pore formed by *cis* complexes.

2. p. 5: The authors show that v-SNARE density and not just copy number, is a crucial parameter for pore formation. This finding should be related to the results of Sharma and Lindau (2018) providing a mechanistic explanation for the requirement of SNARE complex proximity for fusion pore formation.

>> This is an excellent point, and in revision we cited the Sharma and Lindau 2018 paper to support our SNARE density findings.

3. The description/terminology of fusion pore kinetics is not quite clear. The authors use the term of exponential kinetics for pore closure and pore opening but it seems that they actually refer to open and closed time distributions. This is confusing because the open time distribution

actually reflects the kinetics of pore closure and vice versa. This should be clarified as also relevant for the interpretation of the results.

>> We agree, completely, and apologize if this was not clearly stated; in revision, we clarified this point as follows: “CDFs of the closed and open time distributions reflect the kinetics of pore opening and closure, respectively.”.

4. Some recordings show multiple fusion pore openings separated by a closure with long duration, as in Fig. 3, lower trace. Is this the result of the re-opening of a pore formed by the same ND or a new pore formed by a different ND? This question relates to the analysis of pore closure dwell times and fraction of open times.

>> Please see our response to referee #2, point #2b, above, where we address this issue.

5. Related to this question is the pore lifetime in the presence of PIP2. Do all pores eventually close or are they in an indefinite (on experimental time scale) stable open state? This is referred to in Fig 5 for the addition of cd-syb2 and BAPTA but in the preceding experiments it is not clear because open pore lifetimes are presumably limited by brief closing flickers. Is there some “terminal closure” indicated by long lifetime closed state and if so what is the total lifetime of the (somewhat flickering) fusion pores.

>> All pores terminally close after ~90 minutes (but we do not always record for that long). Once the terminal close occurs, there are no openings or flickers. We have appended a typical trace showing closure of a pore here:

Figure legend. Typical recording of a fusion pore formed using ND9_L + syt1 (+ Ca²⁺); this pore eventually closed after ~90 min and did not re-open again, even over extended recording times (several hours). This is typical of all recordings (+/- PIP2 etc.); in all cases pores close after ~90 min.; once the terminal closure occurs no further openings are observed.

We clarified this issue in the revised manuscript by stating that there is a terminal closure in all recordings (when we wait long enough). The revised manuscript now states: “Pore properties did not differ at the beginning, middle, or end of a recording, but all pores closed after ~90 minutes. Once the terminal closure occurred, there were no further openings or flickers.”

6. Fig. 4 and related text: How were the different FRET efficiency states quantified?

>> This issue is clarified in the Methods section as follows: “smFRET data were processed using custom MATLAB scripts. We fitted the FRET histograms with 2 or 3-component Gaussians to derive the FRET efficiency for each state.”

7. The time scale of the smFRET experiments is not fully clear. The authors state an initial 30 min incubation but what was the time scale following Ca²⁺ addition? How does the time scale relate to the waiting time until a fusion pore forms and the pore lifetimes?

>> We incubated with or without Ca²⁺ at the beginning of the experiments. So, it is 30 min following Ca²⁺ addition. This is similar to the waiting time for fusion pore formation in the ND-BLM system (10-30 min). We have clarified this issue in the revised manuscript: “v- and t-SNARE NDs (5 μM) were incubated at room temperature for 30 minutes in the reconstitution buffer supplemented with 0.5 mM EGTA or Ca²⁺. This is similar to the waiting time for fusion pore formation in the ND-BLM system (10-30 min).”

Reviewer #4 (Remarks to the Author):

In all, this is an interesting paper and I am in favor of publishing it if the authors can address my following questions.

>> We thank the referee for her/his positive feedback and suggestions.

Major comments:

1. ND9_L (0.013 Syb2/nm²) vs. ND3_s (0.022 Syb2/nm²): in the absence of syt1/Ca²⁺, why the former yields open pores whereas the latter yields closed pores regarding their similar Syb2 density (Fig. 1b and d).

>> In our manuscript, we argue that both the density, as well as the absolute copy number of SNAREs, are crucial parameters to yield stable open pores. The differences between ND9_L and ND3_s illustrate this point. Although the v-SNARE density is lower in ND9_L versus ND3_s, in the case of ND9_L, the higher copy number of SNAREs serve to hold the pore in a stable open state, whereas in the case of ND3_s, there are not enough copies to yield a stable open state of the pore (please see also Bao et al. (2018) PMID: 29420480, where we show that the number of SNAREs profoundly affects pore stability).

We have clarified these points by stating the following in the revised text: “We next examined ND9_L (Fig. 1a; 0.013 syb2/nm²) lacking syt1; these NDs yielded pores that remained mostly in the open state, but transiently flickered closed (Fig. 1d). In contrast, ND3_s, which has a somewhat higher syb2 density, gave rise to pores that were mostly in closed state but transiently flickered open. We conclude that the higher syb2 copy number in ND9_L serves to hold the pore in a stable open state¹⁶.”

2. Based on Fig. 1e, the author concluded that “inclusion of syt1 in both ND3s and ND9L reduced the occurrence of pores in the absence of Ca²⁺”, (lines 116-117), however, in the trace of Fig. 1b, it seems to me that inclusion of syt1 increase the occurrence of pores in the absence of Ca²⁺. Please clarify this issue.

>> This appears to be a misunderstanding regarding pore open life time versus pore occurrence. When we state “occurrence”, what we mean is that for a given number of trials, what is the fraction of trials in which pore formation occurred? Even when pore formation occurred in relatively few trials (under some experimental conditions), we still show a representative trace of the pores that did form in Fig. 1b, and provide analysis of these pores. We agree that the presence and absence of syt1 (-Ca²⁺) does impact the open life time, as shown - quantitatively - in Figure 2b. However, getting back to the question raised by the

referee, we found that **the occurrence of pores (that is, whether a pore ever formed) was reduced by inclusion of syt1 (in absence of Ca²⁺) in both ND3_s and ND9_L**. In the presence of syt1, the addition of Ca²⁺ always triggered an increase in pore occurrence.

3. In lines 121-122, the conclusion “Hence, the clamping activity of apo-syt1 is strongly dependent on SNARE density and copy number” is confusing and overstated; if so, using higher density of apo-Syt1 while leaving SNARE densities unchanged, the clamping effect will be stronger. Thus, increasing apo-Syt1 copies/densities in ND3s or ND9L should be tested.

>> We carefully considered this point, and opted to conduct what is, in effect, the reciprocal experiment, where we fixed the SNARE density but reduced the syt1 density, to test whether the clamping activity of apo-syt1 depends on the ratio of syt1 to syb2. By reducing the copy number of syt1, we avoid the steric crowding issue (raised by referee #2) on the surface of the nanodiscs that might confound interpretation (please see our response to point #1 from referee #2, above).

The results from these experiments are now included in a new supplementary figure (Supplementary Fig. 2), also shown below:

Supplementary Fig. 2

Supplementary Fig. 2. Titrating the syt1 copy number in NDs.

a, Fraction of trials in which a fusion pore was detected, plotted as % occurrence: ND9_L, in absence (-) and presence (+) of Ca²⁺, and in the absence (-) and presence (+) of the indicated number of reconstituted syt1 molecules, were compared. The total number of measurements obtained under each condition (n) are indicated. b, c, Plots of pore currents and open-time histograms, respectively, as a function of syt1 copy number.

These experiments show that increasing the syt1 copy number in a ND decreases the percent occurrence of ND9_L pores in absence of Ca²⁺, suggesting either defects in docking or increases in the clamping function of syt1. It is difficult to distinguish between these possibilities, as there is no technology available to measure docking in a planar lipid bilayer electrophysiology set-up (we are working to build an entirely new instrument to do this). However, we note that it has been reported that syt1 drives docking (detailed above in response to referee #2 point #1), so increases in syt1 copy number might serve to actually increase, rather than inhibit, docking in our assay. But, again, the correct interpretation cannot be currently distinguished experimentally. To help clarify, we briefly discuss the docking issue in the revised manuscript (as detailed in our response to point 1 from referee #2 above)

Importantly, from the current measurements we found that the size of ND9_L pores decreased as a function of increased syt1 copy number (in absence of Ca²⁺). The effect of syt1 was even more dramatic in our kinetic measurements: increases in syt1 copy number clearly decreased the open life time of fusion pores (in absence of Ca²⁺).

These three quantitative measures demonstrate that we have recapitulated the clamping function of syt1 in the ND-BLM system, in the absence of Ca²⁺. In the presence of Ca²⁺, we observed clear stimulatory effect of syt1 in terms of pore occurrence, current (indicative of pore size) and open life time. In short, these new findings reveal that the clamping activity of syt1 scales with the syb2/syt1 ratio.

4. There is no strong evidence supporting the correlation between “syt1 dithers between the cis and trans” and “fusion pore flickering behavior” stated in lines 210-212. Otherwise, the author should analyze the effect of Syt1 copies/densities in the absence/presence of PI(4,5)P2. More Syt1 is expected to efficiently bind limited SNAREs/SNARE complex thus quenching the flickering, which is important for the correlation.

>> We agree that this was a speculative point, and raised this issue as a topic for a future study. We have removed this statement in the revised manuscript. We plan to address this issue, experimentally, once we are able to bring-in personnel to continue the ND-BLM recordings (as Debasis Das, the first author and BLM specialist, has left our lab to start his own lab in India).

5. Please specify the ND size used in Fig.4, since different NDs (ND3S, ND3L, or ND9L) may make a difference in CC-FRET efficiency.

We apologize for this omission. We now state that they are 13 nm nanodiscs bearing only one v- or t-SNAREs, in the revised manuscript: “ND1_s were used in all experiments”.

6. In line 235, “In short, the ability of syt1 to clamp membrane fusion prior to the Ca²⁺ signal is associated with its ability to drive assembly of trans-SNARE complexes into a more zippered, yet inhibited, state...”. However, the data in Fig. 4 only supports the more zippered model but not the inhibited model. Background and previous results on the clamped/inhibited role of syt1 in the absence of Ca²⁺ (for example, ref. 26 and other refs) should be discussed.

>> In the revised manuscript we now reiterate the previous results regarding the clamping role of syt1, and how this relates to our smFRET findings. Our revised manuscript now states: “We note that our smFRET experiments are in agreement with previous studies, based on force measurements, that examined the assembly of *trans*-SNARE complexes⁵²⁻⁵⁴ (Supplementary Fig. 9a).”

Moreover, we explicitly discuss the clamping activity of apo-syt1, as determined in cell-based experiments, as follows: “These findings are consistent with cell-based studies which indicate that the C2B-domain of apo-syt1 is a potent fusion clamp⁵⁰. This clamping activity appears to be controlled by conformational changes that determine the relative disposition of the tandem C2-domains. Namely, apo-syt1 clamps fusion when the tandem C2-domains of apo-syt1 are askew⁵⁰. Then, upon binding Ca²⁺, the C2-domains reorient and point in the same direction to trigger exocytosis²⁶.”

7. ND-based smFRET should be used to detect the role of NSF/SNAP/ATP in trans-SNARE complex disassembly.

>> We agree that it would be great to detect the disassembly of single *trans*-SNARE complexes using smFRET. However, this experiment is not directly related to our study on fusion pores in Fig. 6, as more than two *trans*-SNARE complexes are required to hold open a fusion pore in the ND-BLM system. Monitoring multiple *trans*-SNARE complexes by smFRET is extremely challenging; we are in the process of trying to develop new approaches, but feel this is beyond the purview of the current study. We also emphasize that the best characterized function of NSF is to disassemble SNAREs in a manner that depends on ATP hydrolysis, and the effects we observe require ATP hydrolysis, as well as both NSF and α -SNAP. We are unaware of any other alternative interpretations of our data, so argue it is reasonable that we are measuring the functional consequences of SNARE disassembly mediate by ATP/ α -SNAP/NSF.

8. Accessory proteins may preclude trans-SNARE complex disassembly by NSF/SNAP/ATP in order to protect the fusion pore formation during exocytosis, for example, HOPS in vacuolar fusion [Song et al, 2017, eLife]; and Munc18/Munc13 in synaptic exocytosis [Ma et al, 2013, Science]. The authors are encouraged to discuss these.

>> This is an excellent point. Please see our response to referee #1 point #5 above, where we state: in the revised manuscript we have included this point in the Discussion: “Interestingly, binding of SM (Sec1-Munc18) proteins in synapses⁵⁸ or the binding of HOPS (homotypic fusion and protein sorting) complex in case of vacuolar fusion⁵⁹ might protect *trans*-SNARE complexes from NSF/ α -SNAP-mediated disassembly⁶⁰. It will be interesting to determine whether these factors affect NSF mediated fusion pore closure in the ND-BLM system described here.”

Minor comments:

1. In line 430, the authors should give a specific hypothesis testing method used in Fig.3c.

>> We apologize for not including this before. Student's T-test was performed to compare the two means. This is now included in the revised manuscript.

2. Chi-square tests were used throughout the manuscript, is it Pearson chi-square test? The author should precisely specify the exact statistical method.

>> Yes, it is Pearson's chi-square test. We have corrected that in our revised manuscript.

3. *In supplementary Fig. 1c, the authors used “Kruskal–Wallis U-test”, please check the words, is it “Kruskal–Wallis H-test”? or “Mann–Whitney U-test”?*

>> >> We apologize for this error; it is “Kruskal–Wallis H-test”. We have corrected this mistake in our revised manuscript.

4. *In lines 616-617, the authors are suggested to rewrite the equation which calculates the FRET efficiency using equation editor.*

>> We have rewritten the equation using the equation editor in the revised manuscript:
“FRET efficiency (E) was calculated using the following equation: $E = \frac{IA - 0.05ID}{ID + IA}$.”

Reviewers' Comments:

Reviewer #1:

Remarks to the Author:

The authors have carefully revised the manuscript to address my previous comments. One more comment: In Figure 4, two donors or two acceptors are labeled on each t- or c-SNARE, which is not the case. The authors may redraw the diagram to show that only one donor or acceptor is attached to each SNARE.

Reviewer #2:

Remarks to the Author:

The authors appropriately responded to the reviewer's comments. As for any interesting manuscript there would still be much more to discuss with the authors. However, the revised manuscript is very clear and provides sufficient and convincing results that deserve to be published in Nature Communications.

Reviewer #1 (Remarks to the Author):

The authors have carefully revised the manuscript to address my previous comments. One more comment: In Figure 4, two donors or two acceptors are labeled on each t- or c-SNARE, which is not the case. The authors may redraw the diagram to show that only one donor or acceptor is attached to each SNARE.

We have redrawn this illustration as suggested.

Reviewer #2 (Remarks to the Author):

The authors appropriately responded to the reviewer's comments. As for any interesting manuscript there would still be much more to discuss with the authors. However, the revised manuscript is very clear and provides sufficient and convincing results that deserve to be published in Nature Communications.

Frederic Pincet

We thank the referee for his insights.

Reviewer #3 (Remarks to the Author):

The manuscript has been improved considerably and my concerns were overall satisfactorily addressed. Regarding my first point (what is the evidence that the NSF acts actually on trans SNARE complexes) the authors provide reasonable arguments. The authors provide a new Supplementary Fig. 12, which is very helpful, illustration of the authors' interpretation. However, when I look at the new Supplementary Fig. 12, I am not sure what the authors consider a trans SNARE complex. Presumably, this is an illustration of the SNARE complex topology in the fusion pore studied here. In this figure, however, it is not evident that there are two distinct membranes, as is a prerequisite of what is traditionally a requirement for the existence of a trans SNARE complex.

We thank the referee for this insightful suggestion. The fusion pore structure in the Supplementary Fig. 12 did not properly summarize our findings. To address this concern, we have further improved this figure to reflect the findings from our earlier study (Bao et al. (2016) Nature SMB) where we showed that the transmembrane domain of syb2 partially lines the ND 'side' of the nascent fusion pore while the TMD of syntaxin 1A partially lines the target membrane 'side' of the fusion pore (in that 2016 study, we used liposomes as the target membrane; these liposomes mimic the BLM used in the current study). We have altered the illustration to reflect these earlier findings, by showing that the v- and t-SNARE transmembrane domains partially line the ND and BLM "sides" of the nascent pore formed between two opposing membranes, as suggested by the referee. So, while the bilayers have some degree of continuity (as our 2016 data revealed that pores are hybrid structures, comprising both lipids and SNARE transmembrane domains), from the perspective of the SNAREs, the TMDs are still anchored on 'opposing' (albeit connected) membranes. We feel this is now much clearer in the further revised figure, where we also now indicate the two leaflets of the bilayer.

The authors claim in their rebuttal that "there is no possible physical mechanism by which the disassembly of fully zippered cis-SNARE complexes (which are defined as having all three

SNAREs fully zippered together all the way through the transmembrane domains (TMD) in the same membrane) can close a pore". The authors' definition of a trans SNARE complex thus appears to rely on the TMDs being not fully zippered. If, however, the topology of Supplementary Fig. 12 keeps the fusion pore open and disassembly by NSF leads to fusion pore closure, then it is not evident to this reviewer why the same could not be possible if the TMDs in Supplementary Fig. 12 would be fully zippered including the TMDs. It would be desirable if these aspects would be addressed when the claim that NSF acts on cis SNARE complexes is discussed.

As indicated in our comments immediately above, our findings (Bao et al. (2016) Nature SMB) indicate that the v- and t-SNARE transmembrane domains partially line the v-SNARE/ND and t-SNARE/liposome/BLM "sides" of the fusion pore, respectively. We argue that if these trans-SNARE complexes are disassembled (by cd-syb or by NSF), then they can no longer hold the pore open, so the pore closes, as supported by our data. To reflect these findings, we have revised the model of fusion pores in Supplementary Fig. 12. In our view, once the SNAREs are allowed to become cis (by expansion of the pore and conversion of the hybrid nascent fusion pore into a larger, purely lipidic fusion 'neck'), their action has become irreversible. In this condition, disassembly would result in un-complexed SNARE proteins diffusing around in the bilayers, but we do not see any possible physical mechanism by which this disassembly can do 'work' on the bilayers to separate the ND from the BLM and to close a pore. In this latter condition, segregation of the membranes would require input of energy and work from other proteins, such as dynamin. We argue that if SNAREs had become cis, NSF would not have closed pores. The observation that NSF did in fact close nascent pores is, in our view, very strong evidence that cis complexes rarely, if ever, form in our system.

Another way to look at this is to flip the question the other way around: how can disassembly of three SNARE proteins, in the same continuous bilayer, cause a fusion pore to close? Since we see no possible mechanism, we have not included this alternative interpretation in our manuscript. If the referee can explain an alternative mechanism, in which disassembly of cis complexes can close a pore, we would be happy to incorporate this alternative model in our revision.

For the quantification of FRET efficiency (my point 6) it should be noted that "in order to determine actual FRET efficiency, one has to determine the correction factor, γ , which accounts for the differences in quantum yield and detection efficiency between the donor and the acceptor" (Roy et al 2008 Nat.Methods 5:507-16). Please either explain if and how this was done or name it FRET ratio "R" rather than FRET efficiency.

We thank the referee for this correction. We have changed the text to state FRET ratio "R" rather than FRET efficiency.

Reviewer #4 (Remarks to the Author):

The authors have addressed my previous concerns in the revised manuscript.

An additional minor comment:

In Fig.4, the authors specified that ND1s were used in all smFRET experiment in the revision. In my view, the authors should explain why ND1s were applied (as responded in point#7) to make it easier to understand by the readers (since multiple ND sizes and protein copy numbers were used in different figures).

To address this issue, we now state, in the body of our manuscript (and not just in our rebuttal letter, that: “Monitoring multiple *trans*-SNARE complexes by smFRET is extremely challenging, so we used ND1 for these experiments.”